# Association Between Upper Respiratory Tract Infections and Parkinson’s Disease in Korean Populations: A Nested Case–Control Study Using a National Health Screening Cohort

**DOI:** 10.3390/brainsci15090939

**Published:** 2025-08-28

**Authors:** Hyuntaek Rim, Hyo Geun Choi, Jee Hye Wee, Joo Hyun Park, Mi Jung Kwon, Ho Suk Kang, Hoang Nguyen, In Bok Chang, Joon Ho Song, Ji Hee Kim

**Affiliations:** 1Department of Neurosurgery, Hallym University College of Medicine, Anyang 14068, Republic of Korea; coldwings@naver.com (H.R.); nscib71@hanmail.net (I.B.C.); song@hallym.or.kr (J.H.S.); 2Department of Otorhinolaryngology-Head and Neck Surgery, Suseoseoul ENT Clinic, Seoul 06349, Republic of Korea; pupen@naver.com; 3MD Analytics, Seoul 06349, Republic of Korea; 4Department of Otorhinolaryngology-Head and Neck Surgery, Hallym University College of Medicine, Anyang 14068, Republic of Korea; weejh07@hanmail.net; 5Department of Otorhinolaryngology-Head and Neck Surgery, Dongguk University Ilsan Hospital, Goyang 10326, Republic of Korea; parkzzu19@naver.com; 6Department of Pathology, Hallym University College of Medicine, Anyang 14068, Republic of Korea; mulank@hanmail.net; 7Division of Gastroenterology, Department of Internal Medicine, Hallym University College of Medicine, Anyang 14068, Republic of Korea; hskang76@hallym.or.kr; 8Department of Chemistry and Biochemistry, University of California San Diego, San Diego, CA 92093, USA; hon005@ucsd.edu

**Keywords:** infection, neurodegenerative disease, neuroinflammation, Parkinson’s disease, pathogenesis, upper respiratory tract infection

## Abstract

Background: Although several epidemiological studies have suggested a potential association between infections and Parkinson’s disease (PD), relatively few have specifically examined the relationship between upper respiratory tract infections (URIs) and PD, apart from coronavirus disease 2019 (COVID-19). Objectives: We investigated whether a history of URI was associated with the diagnosis of PD among Korean individuals aged ≥40 years, using data from the Korean National Health Insurance Service–Health Screening Cohort. Methods: A total of 5844 patients newly diagnosed with PD were identified and matched with 23,376 control participants at a 1:4 ratio based on age, sex, income, and geographical region. Conditional logistic regression analyses were performed to estimate odds ratios (ORs) and 95% confidence intervals (CIs) for PD, adjusting for potential confounders including smoking, alcohol consumption, body mass index, blood pressure, comorbidity scores, blood glucose, and serum cholesterol levels. Results: Overall, no significant association was found between a history of URI and PD when considering a two-year exposure window. However, in the one-year window analysis, individuals with a history of URI had a modestly reduced odds of PD (≥1, ≥2, or ≥3 episodes: (adjusted OR: 0.93, 95% CI: 0.88–0.97, aOR: 0.91, 95% CI: 0.87–0.96 and aOR: 0.92, 95% CI: 0.87–0.98, respectively). Subgroup analyses revealed that the inverse association was more pronounced among women, older adults (≥65 years), and those with higher comorbidity scores. No clear dose–response trend was observed across increasing frequencies of URI diagnoses. Conclusions: Our findings suggest that the apparent protective association between recent URI history and PD is unlikely to be causal and may instead reflect confounding by medication use or reverse causation related to the prodromal phase of PD. These results should therefore be interpreted with caution and regarded as hypothesis-generating. Further prospective studies incorporating detailed prescription data and long-term follow-up are warranted to clarify the role of infections and anti-inflammatory medications in the pathogenesis of PD.

## 1. Introduction

Parkinson’s disease (PD) is a slowly progressive neurodegenerative disorder characterized by the loss of dopaminergic neurons in the midbrain and the accumulation of misfolded alpha-synuclein aggregates, known as Lewy bodies, in various brain regions. It typically presents with motor symptoms such as resting tremor, rigidity, bradykinesia, and postural instability. Non-motor symptoms, arising from dysfunction across multiple neurotransmitter systems in both central and peripheral nervous systems [1], include psychiatric and autonomic disturbances, cognitive impairment, sleep disorders, olfactory dysfunction, and pain [2,3,4]. PD affects approximately 7.5 million people worldwide, with a prevalence that increases markedly with age [5]. Despite extensive research, the precise etiology and pathogenesis of neuronal degeneration in PD remain incompletely understood.

Upper respiratory tract infections (URIs) are among the most common human illnesses, encompassing a broad range of conditions affecting the nasal cavity, sinuses, pharynx, larynx, and large airways [6]. Most acute URIs are viral in origin, primarily caused by rhinoviruses, though they may occasionally result in bacterial complications or spread to adjacent organs [7]. The primary symptoms of URIs are usually mild and include nasal congestion, rhinorrhea, sneezing, sore throat, cough, malaise, fatigue, and low-grade fever. URIs occur globally and annually, with some progressing to more severe manifestations that require hospitalization [8]. Antiviral agents are currently approved only for select pathogens, such as influenza viruses, whereas antibiotic therapy is generally reserved for bacterial complications, such as acute otitis media and sinusitis.

Current understanding of PD pathogenesis implicates several overlapping mechanisms, including aberrant protein handling, oxidative stress, mitochondrial dysfunction, excitotoxicity, and programmed cell death (apoptosis) [9]. Increasing evidence also suggests that neuroinflammatory processes play a critical role in the cascade leading to neuronal degeneration. In both human patients with PD and corresponding animal models, sustained inflammatory responses, T-cell infiltration, and glial cell activation have been consistently observed and are thought to substantially contribute to dopaminergic neuronal loss [10,11].

In this context, both bacterial and viral infections have been identified as potential environmental risk factors for PD, primarily through their ability to induce chronic microglial activation and inflammation [12]. A wide range of bacterial species—including *Helicobacter pylori*, *Escherichia coli*, *Proteus mirabilis*, *Mycobacterium tuberculosis*, *Porphyromonas gingivalis*, *Clostridium difficile*, and *Chlamydia pneumoniae*—which cause infections in the lungs, skin, and gastrointestinal tract, have been associated with the onset and, to a lesser extent, progression of PD [13,14,15,16]. Similarly, several viruses have been implicated in PD development, including influenza virus, coxsackievirus, Japanese encephalitis virus, West Nile Equine Encephalitic virus (WEEV), herpesviruses, hepatitis C virus (HCV), human immunodeficiency virus (HIV), and SARS-CoV-2 [17,18,19].

However, compared to better-studied pathogens such as influenza, relatively few studies have investigated the association between PD and URIs caused by other viral or bacterial agents [20,21,22,23]. In this study, we examined whether individuals with a history of multiple URIs were at increased risk for subsequent PD compared with the general population.

## 2. Methods

### 2.1. Ethics

This study was approved by the Ethics Committee of Hallym University (IRB No.: 2019-10-023) on 22 December 2022. The requirement for written informed consent was waived by the Institutional Review Board. All analyses were conducted in accordance with the relevant guidelines and regulations of the Hallym University Ethics Committee.

### 2.2. Study Design and Population

We conducted a retrospective cohort study using data from the Korean National Health Insurance Service (NHIS)–Health Screening Cohort, covering the period from 1 January 2002 to 31 December 2019. The study population included newly diagnosed patients with PD and matched controls. Among the 514,866 participants with 895,300,177 medical claim codes, individuals diagnosed with PD more than twice during the study period were classified into the PD group (*n* = 9437); while the remaining participants were assigned to the control group (*n* = 505,429).

To exclude potential pre-existing PD cases, participants diagnosed with PD in 2002–2003 (*n* = 641) and those without fasting blood glucose data (*n* = 2) were excluded from the PD group. In the control group, participants with only a single PD diagnosis were excluded (*n* = 2082).

To reduce selection bias and maximize the control sample, each patient with PD was matched to four control participants based on age, sex, income, and geographical region. Controls were randomly ordered and selected sequentially. The index date for each patient with PD was defined as the date of the first PD diagnosis, and the same index date was assigned to their matched controls. A total of 8794 participants in the PD group and 35,176 participants in the control group were included in the final analysis (Figure 1).

### 2.3. Exposure

URIs were defined using specific ICD-10 codes: J00 (acute nasopharyngitis) and all codes from J02 (acute pharyngitis) to J069 (acute upper respiratory tract infection) [24]. Individuals with at least one registered diagnosis of URI during the study period were considered to have a history of URI.

### 2.4. Outcome

PD was defined using ICD-10 code G20 (Parkinson’s disease). To improve diagnostic accuracy, we required at least two separate physician-recorded diagnoses of PD at different clinic visits. Although this approach increases diagnostic validity, it may exclude subclinical or misdiagnosed cases, which could result in underestimation of the true association. We selected exposure windows of 1 year and 2 years prior to PD diagnosis to focus on short-term infection history, as longer windows would likely overlap with the prodromal phase of PD and introduce greater uncertainty in temporal inference.

### 2.5. Covariates

Potential confounding variables were selected based on previous literature and included demographic factors (age, sex, geographical region, and income) and comorbidities (smoking status, alcohol use, obesity, systolic (SBP) and diastolic blood pressure (DBP), fasting blood glucose, and total cholesterol). The Charlson Comorbidity Index (CCI), which quantifies multimorbidity using 17 comorbidities, was used as a continuous variable ranging from 0 (no comorbidities) to 29 (multiple comorbidities).

Age was categorized into 10 groups in 5-year intervals starting from age 40. Income was stratified into five levels, from class 1 (lowest) to class 5 (highest). Geographical regions were classified as urban (Seoul, Busan, Daegu, Incheon, Gwangju, Daejeon, and Ulsan) or rural (Gyeonggi, Gangwon, Chungcheongbuk, Chungcheongnam, Jeollabuk, Jeollanam, Gyeongsangbuk, Gyeongsangnam, and Jeju). Smoking status was categorized as nonsmoker, a past smoker, or current smoker. Alcohol intake was recorded as <1 time/week or ≥1 time/week. Obesity was assessed using body mass index (BMI, kg/m^2^) and categorized using Asia-Pacific criteria from the Western Pacific Regional Office (WPRO) 2000: <18.5 (underweight), 18.5–22.9 (normal), 23–24.9 (overweight), 25–29.9 (obese I), and ≥30 (obese II). SBP (mmHg), DBP (mmHg), fasting blood glucose (mg/dL), and total cholesterol (mg/dL) were also measured.

### 2.6. Statistical Analyses

Standardized differences were used to compare baseline characteristics between the PD and control groups (Table 1). Conditional logistic regression was applied to estimate odds ratios (ORs) and 95% confidence intervals (CIs) for the association between URI and PD, adjusting for matched variables (age, sex, income, and geographical region).

Model 1 was adjusted for smoking status, alcohol use, obesity, and CCI scores. Model 2 was further adjusted for SBP, DBP, fasting blood glucose, and total cholesterol (Table 2). URI history was categorized as ≥1, ≥2, or ≥3 events within the previous year, and ≥1 event within the previous 2 years. Subgroup analyses were performed across all covariates (Table 2, Table 3, Table 4 and Table 5).

All statistical analyses were conducted using SAS version 9.4 (SAS Institute Inc., Cary, NC, USA). A two-tailed *p* value < 0.05 was considered statistically significant.

## 3. Results

The proportions of participants in the PD and control groups were well balanced across age, sex, income, and geographical region, as ensured by the matching process (all standardized differences = 0.00, Table 1). Compared with controls, participants with PD were more likely to be nonsmokers, consume less alcohol, and have higher DBP, fasting glucose levels, and CCI scores (Table 1).

**Table 1 brainsci-15-00939-t001:** General characteristics of participants.

Characteristics	Total Participants		
	PD	Control	Standardized Difference
Age (years old) (*n*, %)			0.00
40–44	5 (0.06)	20 (0.06)	
45–49	66 (0.75)	264 (0.75)	
50–54	224 (2.55)	896 (2.55)	
55–59	498 (5.66)	1992 (5.66)	
60–64	883 (10.04)	3532 (10.04)	
65–69	1347 (15.32)	5388 (15.32)	
70–74	1950 (22.17)	7800 (22.17)	
75–79	2122 (24.13)	8488 (24.13)	
80–84	1293 (14.70)	5172 (14.70)	
85+	406 (4.62)	1624 (4.62)	
Sex (*n*, %)			0.00
Male	4204 (47.81)	16,816 (47.81)	
Female	4590 (52.19)	18,360 (52.19)	
Income (*n*, %)			0.00
1 (lowest)	1624 (18.47)	6496 (18.47)	
2	952 (10.83)	3808 (10.83)	
3	1172 (13.33)	4688 (13.33)	
4	1691 (19.23)	6764 (19.23)	
5 (highest)	3355 (38.15)	13,420 (38.15)	
Geographic region (*n*, %)			0.00
Urban	3326 (37.82)	13,304 (37.82)	
Rural	5468 (62.18)	21,872 (62.18)	
Obesity ^†^ (*n*, %)			0.02
Underweight	318 (3.62)	1283 (3.65)	
Normal	3098 (35.23)	12,521 (35.60)	
Overweight	2308 (26.25)	9289 (26.41)	
Obese I	2772 (31.52)	10,988 (31.24)	
Obese II	298 (3.39)	1095 (3.11)	
Smoking status (*n*, %)			0.09
Nonsmoker	6765 (76.93)	25,888 (73.60)	
Past smoker	1200 (13.65)	5142 (14.62)	
Current smoker	829 (9.43)	4146 (11.79)	
Alcohol use (*n*, %)			0.10
<1 time a week	6243 (70.99)	23,295 (66.22)	
≥1 time a week	2551 (29.01)	11,881 (33.78)	
Systolic blood pressure (*n*, %)			0.00
<120 mmHg	2122 (24.13)	8156 (23.19)	
120–139 mmHg	3967 (45.11)	17,428 (49.55)	
≥140 mmHg	2705 (30.76)	9592 (27.27)	
Diastolic blood pressure (*n*, %)			0.11
<80 mmHg	3651 (41.52)	16,604 (47.20)	
80–89 mmHg	3090 (35.14)	12,529 (35.62)	
≥90 mmHg	2053 (23.35)	6043 (17.18)	
Fasting blood glucose (*n*, %)			0.11
<100 mg/dL	4613 (52.46)	20,128 (57.22)	
100–125 mg/dL	2918 (33.18)	11,078 (31.49)	
≥126 mg/dL	1263 (14.36)	3970 (11.29)	
Total cholesterol (*n*, %)			0.05
<200 mg/dL	5169 (58.78)	19,833 (56.38)	
200–239 mg/dL	2501 (28.44)	10,815 (30.75)	
≥240 mg/dL	1124 (12.78)	4528 (12.87)	
CCI score (*n*, %)			0.29
0	2649 (30.12)	16,827 (47.84)	
1	2030 (23.08)	6867 (19.52)	
≥2	4115 (46.79)	11,482 (32.64)	
The number of URIs (Mean, Standard deviation)			
within 1 year	1.72 (3.89)	1.67 (3.28)	0.01
within 2 years	3.50 (6.75)	3.35 (5.66)	0.02

CCI, Charlson comorbidity index; PD, Parkinson’s disease; URI, upper respiratory tract infection. ^†^ Obesity (BMI, body mass index, kg/m^2^) was categorized as <18.5 (underweight), ≥18.5 to <23 (normal), ≥23 to <25 (overweight), ≥25 to <30 (obese I), and ≥30 (obese II).

The mean number of URIs within 1 year was 1.72 in the PD group and 1.67 in the control group. Within 2 years, the means were 3.5 and 3.35, respectively. These differences were statistically significant (*p* < 0.05 for both comparisons, Table 1).

A significant inverse association was observed between a history of URI within 1 year prior to the index date and the subsequent diagnosis of PD. Specifically, participants with ≥1 URI had an adjusted OR of 0.93 (95% CI: 0.88–0.97, Table 2); those with ≥2 URIs had an OR of 0.91 (95% CI: 0.87–0.96, Table 3); and those with ≥3 URIs had an OR of 0.92 (95% CI: 0.87–0.98, Table 4). Participants who experienced ≥1, ≥2, or ≥3 URIs consistently demonstrated a decreased odds of PD even after various stratifications (Table 2, Table 3 and Table 4).

However, no significant associations were observed in certain subgroups, including participants aged <65 years, males, individuals with high income, rural residents, those who were underweight or of normal weight, past and current smokers, those consuming alcohol ≥1 time per week, individuals with high blood pressure, and those with a CCI score of 0. The remaining subgroups showed a consistent inverse relationship between URI and the development of PD (Table 2, Table 3 and Table 4).

When URI history within 2 years prior to the index date was examined, the inverse association between URI and PD was no longer statistically significant (adjusted OR: 0.97, 95% CI: 0.92–1.01, Table 5). Subgroup analysis also revealed no significant associations, except among participants with a CCI core of 1 (Table 5).

**Table 2 brainsci-15-00939-t002:** Crude and adjusted odds ratios for the association between ≥1 event of URI history within 1 year and PD.

Characteristics	No. of PD	No. of Control	Odds Ratios for PD (95% Confidence Interval)					
	(Exposure/Total, %)	(Exposure/Total, %)	Crude ^†^	*p*-Value	Model 1 ^†,‡^	*p*-Value	Model 2 ^†,§^	*p*-Value
Total (*n* = 43,970)								
No URI	4706/8794 (53.5%)	18,170/35,176 (51.7%)	1		1		1	
≥1 URI	4088/8794 (46.5%)	17,006/35,176 (48.4%)	0.93 (0.89–0.97)	0.002 *	0.92 (0.88–0.96)	0.001 *	0.93 (0.88–0.97)	0.001 *
Age < 65 years old (*n* = 8380)								
No URI	948/1676 (56.6%)	3665/6704 (54.7%)	1		1		1	
≥1 URI	728/1676 (43.4%)	3039/6704 (45.3%)	0.93 (0.83–1.03)	0.163	0.93 (0.88–0.98)	0.005 *	0.90 (0.80–1.00)	0.059
Age ≥ 65 years old (*n* = 35,590)								
No URI	3758/7118 (52.8%)	14,505/28,472 (50.9%)	1		1		1	
≥1 URI	3360/7118 (47.2%)	13,967/28,472 (49.1%)	0.93 (0.88–0.98)	0.005 *	0.94 (0.89–0.99)	0.014 *	0.93 (0.88–0.98)	0.009 *
Men (*n* = 21,020)								
No URI	2308/4204 (54.9%)	9215/16,816 (54.8%)	1		1		1	
≥1 URI	1896/4204 (45.1%)	7601/16,816 (45.2%)	1.00 (0.93–1.07)	0.906	1.00 (0.94–1.08)	0.905	0.99 (0.92–1.06)	0.722
Women (*n* = 22,950)								
No URI	2398/4590 (52.2%)	8955/18,360 (48.8%)	1		1		1	
≥1 URI	2192/4590 (47.8%)	9405/18,360 (51.2%)	0.87 (0.82–0.93)	<0.001 *	0.88 (0.82–0.94)	<0.001 *	0.88 (0.82–0.94)	<0.001 *
Low income (*n* = 18,740)								
No URI	2044/3748 (54.5%)	7757/14,992 (51.7%)	1		1		1	
≥1 URI	1704/3748 (45.5%)	7235/14,992 (48.3%)	0.89 (0.83–0.96)	0.002 *	0.90 (0.84–0.97)	0.005 *	0.88 (0.82–0.95)	0.001 *
High income (*n* = 25,230)								
No URI	2662/5046 (52.8%)	10,413/20,184 (51.6%)	1		1		1	
≥1 URI	2384/5046 (47.3%)	9771/20,184 (48.4%)	0.95 (0.90–1.02)	0.139	0.96 (0.90–1.02)	0.213	0.95 (0.90–1.02)	0.148
Urban residents (*n* = 16,630)								
No URI	1864/3326 (56.0%)	1864/3326 (56.0%)	1		1		1	
≥1 URI	1462/3326 (44.0%)	1462/3326 (44.0%)	0.88 (0.82–0.95)	0.001 *	0.89 (0.82–0.96)	0.002 *	0.88 (0.81–0.95)	0.001 *
Rural residents (*n* = 27,340)								
No URI	2842/5468 (52.0%)	11,132/21,872 (50.9%)	1		1		1	
≥1 URI	2626/5468 (48.0%)	10,740/21,872 (49.1%)	0.96 (0.90–1.02)	0.154	0.97 (0.91–1.02)	0.251	0.95 (0.90–1.01)	0.122
Underweight (*n* = 1601)								
No URI	179/318 (56.3%)	720/1283 (56.1%)	1		1		1	
≥1 URI	139/318 (43.7%)	563/1283 (43.9%)	0.99 (0.78–1.27)	0.956	1.02 (0.79–1.31)	0.871	0.98 (0.76–1.26)	0.876
Normal weight (*n* = 15,619)								
No URI	1659/3098 (53.6%)	6563/12,521 (52.4%)	1		1		1	
≥1 URI	1439/3098 (46.5%)	5958/12,521 (47.6%)	0.96 (0.88–1.03)	0.258	0.96 (0.89–1.04)	0.357	0.96 (0.88–1.04)	0.264
Overweight (*n* = 11,597)								
No URI	1235/2308 (53.5%)	4718/9289 (50.8%)	1		1		1	
≥1 URI	1073/2308 (46.5%)	4571/9289 (49.2%)	0.90 (0.82–0.98)	0.019 *	0.90 (0.82–0.99)	0.025 *	0.91 (0.84–0.99)	0.022 *
Obese (*n* = 15,153)								
No URI	1633/3070 (53.2%)	6169/12,083 (51.1%)	1		1		1	
≥1 URI	1437/3070 (46.8%)	5914/12,083 (48.9%)	0.92 (0.85–0.99)	0.034 *	0.92 (0.85–1.00)	0.049 *	0.91 (0.84–0.99)	0.022 *
Nonsmokers (*n* = 32,653)								
No URI	3585/6765 (53.0%)	13,021/25,888 (50.3%)	1		1		1	
≥1 URI	3180/6765 (47.0%)	12,867/25,888 (49.7%)	0.90 (0.85–0.95)	<0.001 *	0.91 (0.86–0.96)	<0.001 *	0.91 (0.86–0.96)	<0.001 *
Past and current smokers (*n* = 11,317)								
No URI	1121/2029 (55.3%)	5149/9288 (55.4%)	1		1		1	
≥1 URI	908/2029 (44.8%)	4139/9288 (44.6%)	1.01 (0.91–1.11)	0.877	1.01 (0.92–1.12)	0.784	0.99 (0.89–1.09)	0.765
Alcohol use < 1 time a week (*n* = 29,538)								
No URI	3288/6243 (52.7%)	11,775/23,295 (50.6%)	1		1		1	
≥1 URI	2955/6243 (47.3%)	11,520/23,295 (49.5%)	0.92 (0.87–0.97)	0.003 *	0.93 (0.88–0.98)	0.011 *	0.92 (0.87–0.98)	0.005 *
Alcohol use ≥ 1 time a week (*n* = 14,432)								
No URI	1418/2551 (55.6%)	6395/11,881 (53.8%)	1		1		1	
≥1 URI	1133/2551 (44.4%)	5486/11,881 (46.2%)	0.93 (0.85–1.02)	0.105	0.93 (0.85–1.02)	0.105	0.93 (0.85–1.01)	0.090
SBP < 140 mmHg and DBP < 90 mmHg (*n* = 30,119)								
No URI	2937/5669 (51.8%)	12,343/24,450 (50.5%)	1		1		1	
≥1 URI	2732/5669 (48.2%)	12,107/24,450 (49.5%)	0.95 (0.90–1.00)	0.072	0.95 (0.90–1.01)	0.077	0.93 (0.88–0.99)	0.024 *
SBP ≥ 140 mmHg or DBP ≥ 90 mmHg (*n* = 13,851)								
No URI	1769/3125 (56.6%)	5827/10,726 (54.3%)	1		1		1	
≥1 URI	1356/3125 (43.4%)	4899/10,726 (45.7%)	0.91 (0.84–0.99)	0.024 *	0.93 (0.86–1.01)	0.096	0.93 (0.86–1.01)	0.074
Fasting blood glucose < 100 mg/dL (*n* = 24,741)								
No URI	2409/4613 (52.2%)	10,164/20,128 (50.5%)	1		1		1	
≥1 URI	2204/4613 (47.8%)	9964/20,128 (49.5%)	0.93 (0.88–0.99)	0.035 *	0.94 (0.88–1.00)	0.043 *	0.93 (0.87–0.99)	0.021 *
Fasting blood glucose ≥ 100 mg/dL (*n* = 19,229)								
No URI	2297/4181 (54.9%)	8006/15,048 (53.2%)	1		1		1	
≥1 URI	1884/4181 (45.1%)	7042/15,048 (46.8%)	0.93 (0.87–1.00)	0.047 *	0.93 (0.87–1.00)	0.056	0.92 (0.86–0.99)	0.024 *
Total cholesterol < 200 mg/dL (*n* = 25,002)								
No URI	2765/5169 (53.5%)	10,264/19,833 (51.8%)	1		1		1	
≥1 URI	2404/5169 (46.5%)	9569/19,833 (48.3%)	0.93 (0.88–0.99)	0.026 *	0.94 (0.89–1.01)	0.072	0.93 (0.88–0.99)	0.034 *
Total cholesterol ≥ 200 mg/dL (*n* = 18,968)								
No URI	1941/3625 (53.5%)	1941/3625 (53.5%)	1		1		1	
≥1 URI	1684/3625 (46.5%)	1684/3625 (46.5%)	0.92 (0.86–0.99)	0.029 *	0.92 (0.86–0.99)	0.033 *	0.91 (0.85–0.98)	0.016 *
CCI scores = 0 (*n* = 19,476)								
No URI	1404/2649 (53.0%)	8765/16,827 (52.1%)	1		1		1	
≥1 URI	1245/2649 (47.0%)	8062/16,827 (47.9%)	0.96 (0.89–1.05)	0.383	0.97 (0.90–1.06)	0.537	0.97 (0.89–1.05)	0.402
CCI score = 1 (*n* = 8897)								
No URI	1090/2030 (53.7%)	3479/6867 (50.7%)	1		1		1	
≥1 URI	940/2030 (46.3%)	3388/6867 (49.3%)	0.89 (0.80–0.98)	0.016 *	0.89 (0.80–0.98)	0.022 *	0.88 (0.80–0.98)	0.015 *
CCI score ≥ 2 (*n* = 15,597)								
No URI	2212/4115 (53.8%)	5926/11,482 (51.6%)	1		1		1	
≥1 URI	1903/4115 (46.3%)	5556/11,482 (48.4%)	0.92 (0.85–0.99)	0.018 *	0.91 (0.85–0.98)	0.009 *	0.90 (0.84–0.97)	0.005 *

CCI, Charlson Comorbidity Index; DBP, Diastolic blood pressure; PD, Parkinson’s disease; SBP, Systolic blood pressure; URI, upper respiratory tract infection. * Conditional or unconditional logistic regression analysis, significance at *p* < 0.05. ^†^ Stratified model for age, sex, income, and geographic region. ^‡^ Model 1 was adjusted for smoking status, alcohol use, obesity, and CCI scores. ^§^ Model 2 was adjusted for model 1 plus total cholesterol, SBP, DBP, and fasting blood glucose.

**Table 3 brainsci-15-00939-t003:** Crude and adjusted odds ratios for the association between ≥2 events of URI history within 1 year and PD.

Characteristics	No. of PD	No. of Control	Odds Ratios for PD (95% Confidence Interval)					
	(Exposure/Total, %)	(Exposure/Total, %)	Crude ^†^	*p*-Value	Model 1 ^†,‡^	*p*-Value	Model 2 ^†,§^	*p*-Value
Total (*n* = 43,970)								
No URI	6168/8794 (70.1%)	24,047/35,176 (68.4%)	1		1		1	
≥2 URIs	2626/8794 (29.9%)	11,129/35,176 (31.6%)	0.92 (0.87–0.97)	0.001 *	0.91 (0.87–0.96)	0.001 *	0.91 (0.87–0.96)	0.001 *
Age < 65 years old (*n* = 8380)								
No URI	1237/1676 (73.8%)	4858/6704 (72.5%)	1		1		1	
≥2 URIs	439/1676 (26.2%)	1846/6704 (27.5%)	0.93 (0.83–1.05)	0.270	0.93 (0.82–1.05)	0.263	0.90 (0.79–1.02)	0.096
Age ≥ 65 years old (*n* = 35,590)								
No URI	4931/7118 (69.3%)	19,189/28,472 (67.4%)	1		1		1	
≥2 URIs	2187/7118 (30.7%)	9283/28,472 (32.6%)	0.92 (0.87–0.97)	0.002 *	0.92 (0.87–0.98)	0.005 *	0.92 (0.87–0.97)	0.003 *
Men (*n* = 21,020)								
No URI	3011/4204 (71.6%)	11,980/16,816 (71.2%)	1		1		1	
≥2 URIs	1193/4204 (28.4%)	4836/16,816 (28.8%)	0.98 (0.91–1.06)	0.626	0.99 (0.92–1.07)	0.770	0.97 (0.90–1.05)	0.413
Women (*n* = 22,950)								
No URI	3157/4590 (68.8%)	12,067/18,360 (65.7%)	1		1		1	
≥2 URIs	1433/4590 (31.2%)	6293/18,360 (34.3%)	0.87 (0.81–0.93)	<0.001 *	0.88 (0.82–0.94)	<0.001 *	0.87 (0.81–0.94)	<0.001 *
Low income (*n* = 18,740)								
No URI	2653/3748 (70.8%)	10,245/14,992 (68.3%)	1		1		1	
≥2 URIs	1095/3748 (29.2%)	4747/14,992 (31.7%)	0.89 (0.82–0.96)	0.004 *	0.89 (0.83–0.97)	0.006 *	0.87 (0.81–0.95)	0.001 *
High income (*n* = 25,230)								
No URI	3515/5046 (69.7%)	13,802/20,184 (68.4%)	1		1		1	
≥2 URIs	1531/5046 (30.3%)	6382/20,184 (31.6%)	0.94 (0.88–1.01)	0.080	0.95 (0.89–1.01)	0.120	0.94 (0.88–1.01)	0.087
Urban residents (*n* = 16,630)								
No URI	2380/3326 (71.6%)	9219/13,304 (69.3%)	1		1		1	
≥2 URIs	946/3326 (28.4%)	4085/13,304 (30.7%)	0.90 (0.82–0.98)	0.011 *	0.90 (0.83–0.98)	0.016 *	0.89 (0.82–0.97)	0.008 *
Rural residents (*n* = 27,340)								
No URI	3788/5468 (69.3%)	14,828/21,872 (67.8%)	1		1		1	
≥2 URIs	1680/5468 (30.7%)	7044/21,872 (32.2%)	0.93 (0.88–1.00)	0.036 *	0.94 (0.88–1.00)	0.054	0.93 (0.87–0.99)	0.022 *
Underweight (*n* = 1601)								
No URI	224/318 (70.4%)	915/1283 (71.3%)	1		1		1	
≥2 URIs	94/318 (29.6%)	368/1283 (28.7%)	1.04 (0.80–1.37)	0.756	1.07 (0.82–1.41)	0.623	1.04 (0.79–1.37)	0.790
Normal weight (*n* = 15,619)								
No URI	2193/3098 (70.8%)	8702/12,521 (69.5%)	1		1		1	
≥2 URIs	905/3098 (29.2%)	3819/12,521 (30.5%)	0.94 (0.86–1.03)	0.162	0.95 (0.87–1.03)	0.219	0.94 (0.86–1.02)	0.143
Overweight (*n* = 11,597)								
No URI	1615/2308 (70.0%)	6234/9289 (67.1%)	1		1		1	
≥2 URIs	693/2308 (30.0%)	3055/9289 (32.9%)	0.88 (0.79–0.97)	0.009 *	0.88 (0.80–0.97)	0.011 *	0.87 (0.79–0.96)	0.006 *
Obese (*n* = 15,153)								
No URI	2136/3070 (69.6%)	8196/12,083 (67.8%)	1		1		1	
≥2 URIs	934/3070 (30.4%)	3887/12,083 (32.2%)	0.92 (0.85–1.00)	0.064	0.92 (0.85–1.01)	0.066	0.91 (0.84–0.99)	0.036 *
Nonsmokers (*n* = 32,653)								
No URI	4716/6765 (69.7%)	17,404/25,888 (67.2%)	1		1		1	
≥2 URIs	2049/6765 (30.3%)	8484/25,888 (32.8%)	0.89 (0.84–0.94)	<0.001 *	0.90 (0.85–0.96)	0.001 *	0.90 (0.85–0.95)	<0.001 *
Past and current smokers (*n* = 11,317)								
No URI	1452/2029 (71.6%)	6643/9288 (71.5%)	1		1		1	
≥2 URIs	577/2029 (28.4%)	2645/9288 (28.5%)	1.00 (0.90–1.11)	0.971	1.00 (0.90–1.11)	0.994	0.97 (0.87–1.08)	0.567
Alcohol use < 1 time a week (*n* = 29,538)								
No URI	4326/6243 (69.3%)	15,618/23,295 (67.0%)	1		1		1	
≥2 URIs	1917/6243 (30.7%)	7677/23,295 (33.0%)	0.90 (0.85–0.96)	0.001 *	0.91 (0.86–0.97)	0.002 *	0.90 (0.85–0.96)	0.001 *
Alcohol use ≥ 1 time a week (*n* = 14,432)								
No URI	1842/2551 (72.2%)	8429/11,881 (71.0%)	1		1		1	
≥2 URIs	709/2551 (27.8%)	3452/11,881 (29.1%)	0.94 (0.85–1.03)	0.202	0.94 (0.85–1.04)	0.215	0.94 (0.85–1.03)	0.202
SBP < 140 mmHg and DBP < 90 mmHg (*n* = 30,119)								
No URI	3912/5669 (69.0%)	16,491/24,450 (67.5%)	1		1		1	
≥2 URIs	1757/5669 (31.0%)	7959/24,450 (32.6%)	0.93 (0.87–0.99)	0.024 *	0.93 (0.88–0.99)	0.027 *	0.92 (0.86–0.98)	0.007 *
SBP ≥ 140 mmHg or DBP ≥ 90 mmHg (*n* = 13,851)								
No URI	2256/3125 (72.2%)	7556/10,726 (70.5%)	1		1		1	
≥2 URIs	869/3125 (27.8%)	3170/10,726 (29.6%)	0.92 (0.84–1.00)	0.059	0.93 (0.85–1.02)	0.117	0.93 (0.85–1.01)	0.092
Fasting blood glucose < 100 mg/dL (*n* = 24,741)								
No URI	3184/4613 (69.0%)	13,599/20,128 (67.6%)	1		1		1	
≥2 URIs	1429/4613 (31.0%)	6529/20,128 (32.4%)	0.93 (0.87–1.00)	0.056	0.94 (0.87–1.00)	0.063	0.93 (0.86–0.99)	0.034 *
Fasting blood glucose ≥ 100 mg/dL (*n* = 19,229)								
No URI	2984/4181 (71.4%)	10,448/15,048 (69.4%)	1		1		1	
≥2 URIs	1197/4181 (28.6%)	4600/15,048 (30.6%)	0.91 (0.84–0.98)	0.016 *	0.91 (0.84–0.98)	0.016 *	0.90 (0.83–0.97)	0.005 *
Total cholesterol < 200 mg/dL (*n* = 25,002)								
No URI	3614/5169 (69.9%)	13,573/19,833 (68.4%)	1		1		1	
≥2 URIs	1555/5169 (30.1%)	6260/19,833 (31.6%)	0.93 (0.87–1.00)	0.041 *	0.94 (0.88–1.01)	0.092	0.93 (0.87–1.00)	0.041 *
Total cholesterol ≥ 200 mg/dL (*n* = 18,968)								
No URI	2554/3625 (70.5%)	10,474/15,343 (68.3%)	1		1		1	
≥2 URIs	1071/3625 (29.5%)	4869/15,343 (31.7%)	0.90 (0.83–0.98)	0.011 *	0.90 (0.83–0.97)	0.009 *	0.89 (0.82–0.96)	0.004 *
CCI scores = 0 (*n* = 19,476)								
No URI	1853/2649 (70.0%)	11,669/16,827 (69.4%)	1		1		1	
≥2 URIs	796/2649 (30.1%)	5158/16,827 (30.7%)	0.97 (0.89–1.06)	0.532	0.99 (0.90–1.08)	0.787	0.98 (0.9–1.07)	0.664
CCI score = 1 (*n* = 8897)								
No URI	1440/2030 (70.9%)	4590/6867 (66.8%)	1		1		1	
≥2 URIs	590/2030 (29.1%)	2277/6867 (33.2%)	0.83 (0.74–0.92)	0.001 *	0.82 (0.74–0.92)	0.001 *	0.82 (0.73–0.91)	<0.001 *
CCI score ≥ 2 (*n* = 15,597)								
No URI	2875/4115 (69.9%)	7788/11,482 (67.8%)	1		1		1	
≥2 URIs	1240/4115 (30.1%)	3694/11,482 (32.2%)	0.91 (0.84–0.98)	0.016 *	0.90 (0.84–0.98)	0.010 *	0.90 (0.83–0.97)	0.006 *

CCI, Charlson Comorbidity Index; DBP, Diastolic blood pressure; PD, Parkinson’s disease; SBP, Systolic blood pressure; URI, upper respiratory tract infection. * Conditional or unconditional logistic regression analysis, significance at *p* < 0.05. ^†^ Stratified model for age, sex, income, and geographic region. ^‡^ Model 1 was adjusted for smoking status, alcohol use, obesity, and CCI scores. ^§^ Model 2 was adjusted for model 1 plus total cholesterol, SBP, DBP, and fasting blood glucose.

**Table 4 brainsci-15-00939-t004:** Crude and adjusted odds ratios for the association between ≥3 events of URI history within 1 year and PD.

Characteristics	No. of PD	No. of Control	Odds Ratios for PD (95% Confidence Interval)					
	(Exposure/Total, %)	(Exposure/Total, %)	Crude ^†^	*p*-Value	Model 1 ^†,‡^	*p*-Value	Model 2 ^†,§^	*p*-Value
Total (*n* = 43,970)								
No URI	7005/8794 (79.7%)	27,600/35,176 (78.5%)	1		1		1	
≥3 URIs	1789/8794 (20.3%)	7576/35,176 (21.5%)	0.93 (0.88–0.99)	0.015 *	0.92 (0.87–0.98)	0.006 *	0.92 (0.87–0.98)	0.008 *
Age < 65 years old (*n* = 8380)								
No URI	1411/1676 (84.2%)	5541/6704 (82.7%)	1		1		1	
≥3 URIs	265/1676 (15.8%)	1163/6704 (17.4%)	0.89 (0.77–1.04)	0.135	0.90 (0.77–1.04)	0.151	0.87 (0.75–1.01)	0.074
Age ≥ 65 years old (*n* = 35,590)								
No URI	5594/7118 (78.6%)	22,059/28,472 (77.5%)	1		1		1	
≥3 URIs	1524/7118 (21.4%)	6413/28,472 (22.5%)	0.94 (0.88–1.00)	0.044 *	0.94 (0.88–1.00)	0.067	0.94 (0.88–1.00)	0.041 *
Men (*n* = 21,020)								
No URI	3399/4204 (80.9%)	13,544/16,816 (80.5%)	1		1		1	
≥3 URIs	805/4204 (19.2%)	3272/16,816 (19.5%)	0.98 (0.90–1.07)	0.651	0.99 (0.91–1.08)	0.853	0.98 (0.89–1.06)	0.581
Women (*n* = 22,950)								
No URI	3606/4590 (78.6%)	14,056/18,360 (76.6%)	1		1		1	
≥3 URIs	984/4590 (21.4%)	4304/18,360 (23.4%)	0.89 (0.82–0.96)	0.004 *	0.89 (0.83–0.97)	0.005 *	0.89 (0.82–0.96)	0.003 *
Low income (*n* = 18,740)								
No URI	2997/3748 (80.0%)	11,766/14,992 (78.5%)	1		1		1	
≥3 URIs	751/3748 (20.0%)	3226/14,992 (21.5%)	0.91 (0.84–1.00)	0.047 *	0.92 (0.84–1.01)	0.072	0.91 (0.83–0.99)	0.032 *
High income (*n* = 25,230)								
No URI	4008/5046 (79.4%)	15,834/20,184 (78.5%)	1		1		1	
≥3 URIs	1038/5046 (20.6%)	4350/20,184 (21.6%)	0.94 (0.87–1.02)	0.128	0.95 (0.88–1.02)	0.168	0.94 (0.87–1.01)	0.104
Urban residents (*n* = 16,630)								
No URI	2677/3326 (80.5%)	10,464/13,304 (78.7%)	1		1		1	
≥3 URIs	649/3326 (19.5%)	2840/13,304 (21.4%)	0.89 (0.81–0.98)	0.020 *	0.90 (0.82–0.99)	0.031 *	0.89 (0.80–0.98)	0.014 *
Rural residents (*n* = 27,340)								
No URI	4328/5468 (79.2%)	17,136/21,872 (78.4%)	1		1		1	
≥3 URIs	1140/5468 (20.9%)	4736/21,872 (21.7%)	0.95 (0.89–1.02)	0.195	0.96 (0.89–1.03)	0.252	0.95 (0.88–1.02	0.148
Underweight (*n* = 1601)								
No URI	255/318 (80.2%)	1042/1283 (81.2%)	1		1		1	
≥3 URIs	63/318 (19.8%)	241/1283 (18.8%)	1.07 (0.78–1.46)	0.676	1.09 (0.80–1.49)	0.585	1.05 (0.77–1.44)	0.769
Normal weight (*n* = 15,619)								
No URI	2496/3098 (80.6%)	9934/12,521 (79.3%)	1		1		1	
≥3 URIs	602/3098 (19.4%)	2587/12,521 (20.7%)	0.93 (0.84–1.02)	0.129	0.94 (0.85–1.03)	0.189	0.92 (0.84–1.02)	0.122
Overweight (*n* = 11,597)								
No URI	1833/2308 (79.4%)	7155/9289 (77.0%)	1		1		1	
≥3 URIs	475/2308 (20.6%)	2134/9289 (23.0%)	0.87 (0.78–0.97)	0.014 *	0.87 (0.78–0.97)	0.015 *	0.86 (0.77–0.96)	0.010 *
Obese (*n* = 15,153)								
No URI	2421/3070 (78.9%)	9469/12,083 (78.4%)	1		1		1	
≥3 URIs	649/3070 (21.1%)	2614/12,083 (21.6%)	0.97 (0.88–1.07)	0.554	0.97 (0.88–1.07)	0.550	0.96 (0.87–1.06)	0.143
Nonsmokers (*n* = 32,653)								
No URI	5355/6765 (79.2%)	20,129/25,888 (77.8%)	1		1		1	
≥3 URIs	1410/6765 (20.8%)	5759/25,888 (22.3%)	0.92 (0.86–0.98)	0.013 *	0.93 (0.87–0.99)	0.032 *	0.92 (0.86–0.99)	0.017 *
Past and current smokers (*n* = 11,317)								
No URI	1650/2029 (81.3%)	7471/9288 (80.4%)	1		1		1	
≥3 URIs	379/2029 (18.7%)	1817/9288 (19.6%)	0.94 (0.84–1.07)	0.362	0.95 (0.84–1.07)	0.397	0.92 (0.81–1.05)	0.208
Alcohol use < 1 time a week (*n* = 29,538)								
No URI	4945/6243 (79.2%)	18,091/23,295 (77.7%)	1		1		1	
≥3 URIs	1298/6243 (20.8%)	5204/23,295 (22.3%)	0.91 (0.85–0.98)	0.009 *	0.92 (0.86–0.99)	0.022 *	0.91 (0.85–0.98)	0.008 *
Alcohol use ≥ 1 time a week (*n* = 14,432)								
No URI	2060/2551 (80.8%)	9509/11,881 (80.0%)	1		1		1	
≥3 URIs	491/2551 (19.3%)	2372/11,881 (20.0%)	0.96 (0.86–1.06)	0.410	0.96 (0.86–1.07)	0.419	0.95 (0.85–1.06)	0.389
SBP < 140 mmHg and DBP < 90 mmHg (*n* = 30,119)								
No URI	4471/5669 (78.9%)	19,009/24,450 (77.8%)	1		1		1	
≥3 URIs	1198/5669 (21.1%)	5441/24,450 (22.3%)	0.94 (0.87–1.00)	0.067	0.94 (0.87–1.00)	0.065	0.92 (0.86–0.99)	0.021 *
SBP ≥ 140 mmHg or DBP ≥ 90 mmHg (*n* = 13,851)								
No URI	2534/3125 (81.1%)	8591/10,726 (80.1%)	1		1		1	
≥3 URIs	591/3125 (18.9%)	2135/10,726 (19.9%)	0.94 (0.85–1.04)	0.219	0.96 (0.87–1.06)	0.426	0.95 (0.86–1.06)	0.351
Fasting blood glucose < 100 mg/dL (*n* = 24,741)								
No URI	3655/4613 (79.2%)	15,699/20,128 (78.0%)	1		1			
≥3 URIs	958/4613 (20.8%)	4429/20,128 (22.0%)	0.93 (0.86–1.01)	0.066	0.93 (0.86–1.01)	0.083	0.92 (0.85–1.00)	0.048 *
Fasting blood glucose ≥ 100 mg/dL (*n* = 19,229)								
No URI	3350/4181 (80.1%)	11,901/15,048 (79.1%)	1		1		1	
≥3 URIs	831/4181 (19.9%)	3147/15,048 (20.9%)	0.94 (0.86–1.02)	0.143	0.94 (0.86–1.02)	0.152	0.92 (0.85–1.01)	0.067
Total cholesterol < 200 mg/dL (*n* = 25,002)								
No URI	4130/5169 (79.9%)	15,669/19,833 (79.0%)	1		1		1	
≥3 URIs	1039/5169 (20.1%)	4164/19,833 (21.0%)	0.95 (0.88–1.02)	0.158	0.96 (0.89–1.04)	0.295	0.95 (0.88–1.02)	0.158
Total cholesterol ≥ 200 mg/dL (*n* = 18,968)								
No URI	2875/3625 (79.3%)	11,931/15,343 (77.8%)	1		1		1	
≥3 URIs	750/3625 (20.7%)	3412/15,343 (22.2%)	0.91 (0.83–1.00)	0.043 *	0.90 (0.83–0.99)	0.028 *	0.90 (0.82–0.98)	0.018 *
CCI scores = 0 (*n* = 19,476)								
No URI	2104/2649 (79.4%)	13,397/16,827 (79.6%)	1		1		1	
≥3 URIs	2104/2649 (79.4%)	13,397/16,827 (79.6%)	1.01 (0.91–1.12)	0.821	1.03 (0.93–1.14)	0.524	1.03 (0.93–1.14)	0.587
CCI score = 1 (*n* = 8897)								
No URI	1616/2030 (79.6%)	5307/6867 (77.3%)	1		1		1	
≥3 URIs	414/2030 (20.4%)	1560/6867 (22.7%)	0.87 (0.77–0.98)	0.027 *	0.88 (0.78–0.99)	0.034 *	0.87 (0.77–0.99)	0.029 *
CCI score ≥ 2 (*n* = 15,597)								
No URI	3285/4115 (79.8%)	8896/11,482 (77.5%)	1		1		1	
≥3 URIs	830/4115 (20.2%)	2586/11,482 (22.5%)	0.87 (0.80–0.95)	0.002 *	0.86 (0.79–0.94)	0.001 *	0.86 (0.78–0.94)	0.001 *

CCI, Charlson Comorbidity Index; DBP, Diastolic blood pressure; PD, Parkinson’s disease; SBP, Systolic blood pressure; URI, upper respiratory tract infection. * Conditional or unconditional logistic regression analysis, significance at *p* < 0.05. ^†^ Stratified model for age, sex, income, and geographic region. ^‡^ Model 1 was adjusted for smoking status, alcohol use, obesity, and CCI scores. ^§^ Model 2 was adjusted for model 1 plus total cholesterol, SBP, DBP, and fasting blood glucose.

**Table 5 brainsci-15-00939-t005:** Crude and adjusted odds ratios for the association between ≥1 event of URI history within 2 years and PD.

Characteristics	No. of PD	No. of Control	Odds Ratios for PD (95% Confidence Interval)					
	(Exposure/Total, %)	(Exposure/Total, %)	Crude ^†^	*p*-Value	Model 1 ^†,‡^	*p*-Value	Model 2 ^†,§^	*p*-Value
Total (*n* = 43,970)								
No URI	3149/8794 (35.8%)	12,374/35,176 (35.2%)	1		1		1	
≥1 URI	5645/8794 (64.2%)	22,802/35,176 (64.8%)	0.97 (0.93–1.02)	0.267	0.96 (0.91–1.01)	0.093	0.97 (0.92–1.01)	0.162
Age < 65 years old (*n* = 8380)								
No URI	643/1676 (38.4%)	2556/6704 (38.1%)	1		1		1	
≥1 URI	1033/1676 (61.6%)	4148/6704 (61.9%)	0.99 (0.89–1.11)	0.857	1.00 (0.89–1.11)	0.945	0.96 (0.85–1.07)	0.449
Age ≥ 65 years old (*n* = 35,590)								
No URI	2506/7118 (35.2%)	9818/28,472 (34.5%)	1		1		1	
≥1 URI	4612/7118 (64.8%)	18,654/28,472 (65.5%)	0.97 (0.92–1.02)	0.250	0.98 (0.93–1.03)	0.425	0.97 (0.92–1.02)	0.256
Men (*n* = 21,020)								
No URI	1608/4204 (38.3%)	6475/16,816 (38.5%)	1		1		1	
≥1 URI	2596/4204 (61.8%)	10,341/16,816 (61.5%)	1.01 (0.94–1.08)	0.761	1.02 (0.95–1.10)	0.526	1.00 (0.93–1.07)	0.986
Women (*n* = 22,950)								
No URI	1541/4590 (33.6%)	5899/18,360 (32.1%)	1		1		1	
≥1 URI	3049/4590 (66.4%)	12,461/18,360 (67.9%)	0.94 (0.87–1.00)	0.062	0.94 (0.88–1.01)	0.092	0.94 (0.87–1.00)	0.061
Low income (*n* = 18,740)								
No URI	1373/3748 (36.6%)	5362/14,992 (35.8%)	1		1		1	
≥1 URI	2375/3748 (63.4%)	9630/14,992 (64.2%)	0.96 (0.89–1.04)	0.321	0.97 (0.90–1.05)	0.477	0.95 (0.88–1.02)	0.153
High income (*n* = 25,230)								
No URI	1776/5046 (35.2%)	7012/20,184 (34.7%)	1		1		1	
≥1 URI	3270/5046 (64.8%)	13,172/20,184 (65.3%)	0.98 (0.92–1.05)	0.543	0.99 (0.93–1.06)	0.734	0.98 (0.92–1.04)	0.488
Urban residents (*n* = 16,630)								
No URI	1264/3326 (38.0%)	4897/13,304 (36.8%)	1		1		1	
≥1 URI	2062/3326 (62.0%)	8407/13,304 (63.2%)	0.95 (0.88–1.03)	0.202	0.95 (0.88–1.03)	0.235	0.94 (0.87–1.02)	0.133
Rural residents (*n* = 27,340)								
No URI	1885/5468 (34.5%)	7477/21,872 (34.2%)	1		1		1	
≥1 URI	3583/5468 (65.5%)	14,395/21,872 (65.8%)	0.99 (0.93–1.05)	0.688	1.00 (0.94–1.06)	0.998	0.98 (0.92–1.05)	0.553
Underweight (*n* = 1601)								
No URI	135/318 (42.5%)	505/1283 (39.4%)	1		1		1	
≥1 URI	183/318 (57.6%)	778/1283 (60.6%)	0.88 (0.69–1.13)	0.314	0.90 (0.70–1.16)	0.437	0.88 (0.68–1.13)	0.309
Normal weight (*n* = 15,619)								
No URI	1083/3098 (35.0%)	4525/12,521 (36.1%)	1		1		1	
≥1 URI	2015/3098 (65.0%)	7996/12,521 (63.9%)	1.05 (0.97–1.14)	0.220	1.06 (0.98–1.15)	0.153	1.05 (0.96–1.14)	0.103
Overweight (*n* = 11,597)								
No URI	823/2308 (35.7%)	3160/9289 (34.0%)	1		1		1	
≥1 URI	1485/2308 (64.3%)	6129/9289 (66.0%)	0.93 (0.85–1.02)	0.138	0.93 (0.85–1.03)	0.166	0.92 (0.84–1.02)	0.103
Obese (*n* = 15,153)								
No URI	1108/3070 (36.1%)	4184/12,083 (34.6%)	1		1		1	
≥1 URI	1962/3070 (63.9%)	7899/12,083 (65.4%)	0.94 (0.86–1.02)	0.129	0.94 (0.87–1.03)	0.178	0.93 (0.85–1.01)	0.077
Nonsmokers (*n* = 32,653)								
No URI	2366/6765 (35.0%)	8689/25,888 (33.6%)	1		1		1	
URI ≥ 1	4399/6765 (65.0%)	17,199/25,888 (66.4%)	0.94 (0.89–0.99)	0.029 *	0.95 (0.90–1.01)	0.086	0.95 (0.89–1.00)	0.054
Past and current smokers (*n* = 11,317)								
No URI	783/2029 (38.6%)	3685/9288 (39.7%)	1		1		1	
≥1 URI	1246/2029 (61.4%)	5603/9288 (60.3%)	1.05 (0.95–1.15)	0.367	1.06 (0.96–1.17)	0.285	1.02 (0.92–1.13)	0.692
Alcohol use < 1 time a week (*n* = 29,538)								
No URI	2190/6243 (35.1%)	7914/23,295 (34.0%)	1		1		1	
≥1 URI	4053/6243 (64.9%)	15,381/23,295 (66.0%)	0.95 (0.90–1.01)	0.102	0.97 (0.91–1.03)	0.255	0.95 (0.90–1.01)	0.109
Alcohol use ≥ 1 time a week (*n* = 14,432)								
No URI	959/2551 (37.6%)	4460/11,881 (37.5%)	1		1		1	
≥1 URI	1592/2551 (62.4%)	7421/11,881 (62.5%)	0.95 (0.90–1.01)	0.109	1.00 (0.92–1.10)	0.969	0.99 (0.90–1.08)	0.822
SBP < 140 mmHg and DBP < 90 mmHg (*n* = 30,119)								
No URI	1902/5669 (33.6%)	8295/24,450 (33.9%)	1		1		1	
≥1 URI	3767/5669 (66.5%)	16,155/24,450 (66.1%)	1.02 (0.96–1.08)	0.591	1.02 (0.96–1.08)	0.541	1.00 (0.94–1.06)	0.989
SBP ≥ 140 mmHg or DBP ≥ 90 mmHg (*n* = 13,851)								
No URI	1247/3125 (39.9%)	4079/10,726 (38.0%)	1		1		1	
≥1 URI	1878/3125 (60.1%)	6647/10,726 (62.0%)	0.92 (0.85–1.00)	0.058	0.95 (0.87–1.03)	0.186	0.93 (0.86–1.01)	0.094
Fasting blood glucose < 100 mg/dL (*n* = 24,741)								
No URI	1587/4613 (34.4%)	6898/20,128 (34.3%)	1		1		1	
≥1 URI	3026/4613 (65.6%)	13,230/20,128 (65.7%)	0.99 (0.93–1.06)	0.865	1.00 (0.93–1.07)	0.961	0.99 (0.92–1.06)	0.677
Fasting blood glucose ≥ 100 mg/dL (*n* = 19,229)								
No URI	1562/4181 (37.4%)	5476/15,048 (36.4%)	1		1		1	
≥1 URI	2619/4181 (62.6%)	9572/15,048 (63.6%)	0.96 (0.89–1.03)	0.248	0.96 (0.90–1.04)	0.309	0.94 (0.88–1.01)	0.101
Total cholesterol < 200 mg/dL (*n* = 25,002)								
No URI	1861/5169 (36.0%)	7019/19,833 (35.4%)	1		1		1	
≥1 URI	3308/5169 (64.0%)	12,814/19,833 (64.6%)	0.97 (0.91–1.04)	0.411	0.99 (0.93–1.06)	0.761	0.97 (0.91–1.04)	0.419
Total cholesterol ≥ 200 mg/dL (*n* = 18,968)								
No URI	1288/3625 (35.5%)	5355/15,343 (34.9%)	1		1		1	
≥1 URI	2337/3625 (64.5%)	9988/15,343 (65.1%)	0.97 (0.90–1.05)	0.474	0.97 (0.90–1.05)	0.443	0.95 (0.88–1.03)	0.234
CCI scores = 0 (*n* = 19,476)								
No URI	1288/3625 (35.5%)	5355/15,343 (34.9%)	1		1		1	
≥1 URI	2337/3625 (64.5%)	9988/15,343 (65.1%)	1.03 (0.94–1.12)	0.566	1.04 (0.95–1.13)	0.394	0.96 (0.94–0.99)	0.559
CCI score = 1 (*n* = 8897)								
No URI	735/2030 (36.2%)	2326/6867 (33.9%)	1		1		1	
≥1 URI	1295/2030 (63.8%)	4541/6867 (66.1%)	0.90 (0.81–1.00)	0.052	0.91 (0.82–1.01)	0.075	0.90 (0.81–1.00)	0.045 *
CCI score ≥ 2 (*n* = 15,597)								
No URI	1477/4115 (35.9%)	3999/11,482 (34.8%)	1		1		1	
≥1 URI	2638/4115 (64.1%)	7483/11,482 (65.2%)	0.95 (0.89–1.03)	0.218	0.94 (0.87–1.02)	0.124	0.93 (0.87–1.01)	0.074

CCI, Charlson Comorbidity Index; DBP, Diastolic blood pressure; PD, Parkinson’s disease; SBP, Systolic blood pressure; URI, upper respiratory tract infection. * Conditional or unconditional logistic regression analysis, significance at *p* < 0.05. ^†^ Stratified model for age, sex, income, and geographic region. ^‡^ Model 1 was adjusted for smoking status, alcohol use, obesity, and CCI scores. ^§^ Model 2 was adjusted for model 1 plus total cholesterol, SBP, DBP, and fasting blood glucose.

## 4. Discussion

This study identified a modest inverse association between the occurrence of URI within 1 year and the subsequent diagnosis of PD. However, when the observation period was extended to 2 years, the association was no longer statistically significant. These findings were generally consistent across stratified analyses, suggesting that the observed relationship is relatively robust.

Historical evidence linking infections to PD dates back to the H1N1 influenza pandemic in 1918, which was associated with cases of encephalitis lethargica and post-encephalic parkinsonism [25]. Since then, both epidemiological and experimental studies have suggested that systemic inflammation caused by infections may play a role in the etiology and progression of PD [26]. Elevated levels of systemic inflammatory markers, such as interleukin-1β, interleukin-6, tumor necrosis factor-alpha, and C-reactive protein, have been reported in both patients with PD and animal models [26,27].

A cohort study evaluating antibody titers against common pathogens found higher seropositivity to cytomegalovirus (CMV), Epstein–Barr virus (EBV), herpesvirus, *Borrelia burgdorferi*, *Chlamydophila pneumoniae*, and *Helicobacter pylori* in patients with PD compared with healthy controls [28]. A recent meta-analysis of cohort and case–control studies further confirmed that infections with *H. pylori*, *C. pneumoniae*, HCV, or *Malassezia* yeast were positively associated with the risk of PD [5].

Infections with other pathogenic microorganisms, including hepatitis B virus, influenza virus, measles, varicella-zoster virus, pertussis, scarlet fever, rheumatic fever, and diphtheria, have also been increasingly recognized as potential risk factors for PD [22,29]. In contrast, certain infections may not increase PD risk and may even exert a protective effect. A population-based case–control study reported an inverse association between PD and childhood infections such as chickenpox, mumps, and measles [22].

Whereas most prior studies have focused on specific pathogens such as influenza, HCV, and *H. pylori*, few have examined the broader category of URI in relation to PD risk [5,23,30,31,32,33,34,35,36,37,38,39,40,41]. Moreover, although many observational studies have explored the association between early- and mid-life infections and PD development, the short-term effects of recent infections have not been well characterized. Interest in the potential short- and long-term cognitive and neurological sequelae of coronavirus disease 2019 (COVID-19) has further emphasized the importance of studying the neuroinflammatory consequences of recent viral exposure.

In this nationwide nested case–control study using physician-coded diagnoses, we found statistically significant inverse association between a history of URIs and PD diagnosis within a 1-year period. Specifically, those with ≥1, ≥2, or ≥3 episodes of URI had 7% (95% CI: 0.88–0.97), 9% (95% CI: 0.87–0.96), and 8% (95% CI: 0.87–0.98) lower odds of developing PD, respectively. However, no significant association was observed when URI exposure was assessed over a 2-year window.

These findings suggest that the protective effect may not be driven by specific pathogens but rather by broader immunological mechanisms, such as systemic inflammation or treatment responses. Although secondary bacterial infections and some viral infections may necessitate antimicrobial or antiviral treatment, the management of URIs is generally supportive. In outpatient settings, clinicians typically prescribe analgesics and antipyretics to alleviate symptoms resulting from local and systemic inflammatory responses. Consequently, one possible explanation for the reduced diagnosis of PD among individuals with recent URIs may involve the protective role of anti-inflammatory medications commonly used to treat such infections.

Indeed, several animal studies have reported consistent findings and described multiple mechanisms through which non-steroidal anti-inflammatory drugs (NSAIDs)—commonly prescribed for pain, fever, and inflammation—may confer neuroprotection in PD. These mechanisms include reducing dopaminergic neuronal loss by downregulating gene-1 expression, which may be associated with the suppression of microglial inactivation. NSAIDs have also been shown to attenuate nuclear factor kappa B (NF-κB) activity, enhance reactive oxygen species (ROS) scavenging, decrease superoxide anion generation, and limit the depletion of dopamine metabolites such as 3,4-dihydroxyphenylacetic acid and homovanillic acid, all of which may contribute to neuroprotection [42,43,44].

Despite accumulating evidence supporting the neuroprotective potential of NSAIDs [45,46,47,48,49], findings remain inconsistent. Several animal studies have shown that COX-2 inhibitors fail to exert neuroprotective effects or reduce neuronal cell death [50]. Similarly, epidemiological studies have yielded conflicting results, reporting both decreased [50] and increased risks of PD associated with NSAID use [51]. In principle, the Korean NHIS claims database contains comprehensive prescription records, which could enable direct evaluation of this hypothesis. However, extraction and validation of medication data require at least 6 to 12 months, and such analyses are therefore beyond the scope of the present study. We acknowledge this limitation and highlight that the observed inverse association may reflect medication effects rather than infections per se. Future studies incorporating prescription records and prospective data are warranted to clarify the neuroinflammatory and neuroprotective roles of these agents in PD pathogenesis.

The subgroup results provide additional insights. The absence of an association in participants with a CCI score of 0 suggests that the effect is not universal but may depend on comorbid health status. The stronger association observed in women may reflect sex-specific immune responses or differences in healthcare-seeking behavior. Associations limited to overweight and obese individuals may indicate an interaction between acute infection and the chronic low-grade inflammation characteristic of obesity. Finally, the restriction of the association to older adults (≥65 years) may be related to immunosenescence, which alters immune responses to infections. Importantly, we observed no clear monotonic dose–response pattern (≥1, ≥2, ≥3 URIs), which further weakens any argument for causality.

The major strength of this study is the use of data from the Korean NHIS, which encompasses nearly the entire Korean population. This enabled a nationwide study design with a large sample size and near-complete follow-up. Moreover, as a population-based study conducted within a single country, it is likely that diagnostic coding practices were consistently applied over time. The use of prospectively recorded diagnoses for both URI and PD also minimizes the risk of selection and measurement bias that may arise in observational studies relying on retrospective data.

Several limitations should be considered. First, reliance on ICD-10 codes may have led to misclassification of both URIs and PD. Although we required at least two PD diagnoses to improve specificity, this may have excluded patients with only one diagnosis, some of whom could have had subclinical or early PD. Second, the retrospective design precludes causal inference. In particular, reverse causation must be strongly considered. The prodromal phase of PD may begin 10–20 years before clinical diagnosis [52] and is characterized by non-motor features such as autonomic dysfucntion, impaired cough reflex, and altered healthcare-seeking behaviors. These prodromal features could reduce the likelihood of URI diagnoses before PD onset [53]. Notably the inverse association was observed only within the 1-year window but not at 2 years, strongly suggesting that our finding reflects early disease processes rather than a causal protective effect of infections. Third, residual and unmeasured confounding (e.g., genetic susceptibility, lifestyle factors) may have influenced the results. Nevertheless, our study adjusted for key known risk factors for PD, including age and smoking, and further accounted for additional factors such as alcohol use, blood pressure, BMI, cholesterol, and fasting blood glucose, which were obtained during standardized health screenings. Fourth, surveillance bias and differences in health-seeking behavior could also partly explain the observed associations, as individuals who seek more medical care may have higher chances of receiving both URI and PD diagnoses. Fifth, although we observed a modest inverse association within 1 year, the effect size was small and not consistent across all subgroups, underscoring the need for cautious interpretation. Lastly, we were unable to conduct formal statistical interaction testing (e.g., URI sex or URI age group terms) due to dataset constraints, which represents an additional limitation. The absence of consistent dose–response trends across increasing URI frequency also suggests caution in interpreting the results. Future prospective cohort studies with detailed symptom tracking are warranted to clarify the temporal relationship and potential biological mechanisms linking infections to PD.

## 5. Conclusions

Our study provides epidemiological evidence of a potential short-term inverse association between URI history and PD diagnosis. This association does not imply causality, and its clinical significance may be limited. Nonetheless, these findings may help generate new hypothesis and underscore the importance of further research exploring the role of infections and inflammation in PD pathogenesis. Future studies incorporating prescription data and prospective designs are needed to clarify the relationship between infections, medications, and PD.

## Figures and Tables

**Figure 1 brainsci-15-00939-f001:**
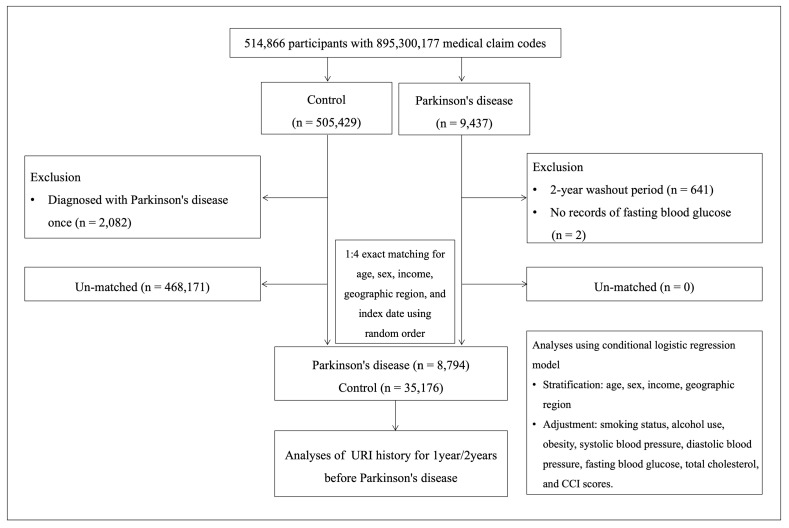
Flowchart illustrating the participant selection process. Among the 514,866 participants, 8794 participants with Parkinson’s disease were matched with 35,176 controls based on age, sex, income, and geographic region.

## Data Availability

The data used for this study are available from the Korean National Health Insurance Sharing Service (https://nhiss.nhis.or.kr) subject to their requirements and fees. For data requests for this study, please contact the corresponding author (kimjihee.ns@gmail.com).

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
