# Peer review of "Association Between Upper Respiratory Tract Infections and Parkinson’s Disease in Korean Populations: A Nested Case–Control Study Using a National Health Screening Cohort"

_brainsci, 2025, doi:10.3390/brainsci15090939_

Round 1
Reviewer 1 Report
Comments and Suggestions for Authors
The authors have conducted a meta-analysis study to investigate the association between upper respiratory tract infections and Parkinson’s Disease (PD) in the Korean population. The study is well-designed, and appropriate statistical analysis is used to conclude the findings. However, I have some major concerns:
- First and foremost, although some reports suggest connections between certain viral or bacterial infections and glial activation, the overall evidence about these links remains inconclusive regarding the potential causes of PD. Therefore, associating upper respiratory tract infections and PD is, I believe, an overestimation.
- The results obtained from this study were predominantly inconclusive (as expected), and the conclusions derived from these results appear to be overstated.
Author Response
The authors have conducted a meta-analysis study to investigate the association between upper respiratory tract infections and Parkinson’s Disease (PD) in the Korean population. The study is well-designed, and appropriate statistical analysis is used to conclude the findings. However, I have some major concerns:
First and foremost, although some reports suggest connections between certain viral or bacterial infections and glial activation, the overall evidence about these links remains inconclusive regarding the potential causes of PD. Therefore, associating upper respiratory tract infections and PD is, I believe, an overestimation. The results obtained from this study were predominantly inconclusive (as expected), and the conclusions derived from these results appear to be overstated.
|
Response: We sincerely appreciate the reviewer’s thoughtful comments. We fully agree that current evidence linking infections and glial activation to the pathogenesis of Parkinson’s disease (PD) is inconclusive, and therefore, any direct causal interpretation should be made with caution. In our revised manuscript, we have clarified that our study does not claim a causal relationship between upper respiratory tract infections (URIs) and PD. Instead, we emphasize that the observed modest inverse association within 1 year Is only an epidemiological finding that warrants further mechanistic research. Furthermore, we have revised the conclusion to avoid overstatement. We now describe our results as indicating a potential short-term inverse association between recent URI history and PD diagnosis, added explicit statements in the Discussion and Conclusion sections highlighting the limitations of observational design, possible reverse causation, and small effect size. We believe these revisions address the reviewer’s concern that our conclusions were overstated.
Previous abstract: “These findings suggest a potential inverse relationship between recent URI and PD development, warranting further investigation into underlying mechanisms.” Revised abstract (pages 1-2, lines 41-108): “Our findings suggest that the apparent protective association between recent URI history and PD is unlikely to be causal and may instead reflect confounding by medication use or reverse causation related to the prodromal phase of PD. These results should therefore be interpreted with caution and regarded as hypothesis-generating. Further prospective studies incorporating detailed prescription data and long-term follow-up are warranted to clarify the role of infections and anti-inflammatory medications in the pathogenesis of PD.”
Previous conclusion: “Although the clinical significance of this finding may be limited due to the small effect sizes, our study provides evidence that adults with a recent history of multiple URIs within the prior year may have a slightly lower likelihood of receiving a PD diagnosis. Further studies are needed to investigate the pathological mechanisms linking infection and its treatment to PD development.” Revised conclusion (page 30, lines 177-183): “Our study provides epidemiological evidence of a potential short-term inverse association between URI history and PD diagnosis. This association does not imply causality, and its clinical significance may be limited. Nonetheless, these findings may help generate new hypothesis and underscore the importance of further research exploring the role of infections and inflammation in PD pathogenesis. Future studies incorporating prescription data and prospective designs are needed to clarify the relationship between infections, medications, and PD.” |
Reviewer 2 Report
Comments and Suggestions for Authors
First of all i want thank to the authors for this great work.The study's reliance on ICD-10 codes may have led to misclassification of both upper respiratory tract infections (URTIs) and Parkinson's disease (PD), which could have impacted the accuracy of the findings. Additionally, the retrospective design limits the ability to establish a causal relationship between URTIs and PD. Exclusion criteria, such as excluding patients with only one diagnosis of PD, may miss subclinical or misdiagnosed cases, thus underestimating the true association. To strengthen the methodology, authors should provide more details on how repeat diagnoses were confirmed as PD, such as whether the diagnoses were confirmed by a specialist or based on additional clinical criteria. Furthermore, the rationale for limiting URTI diagnoses to the year or two preceding the PD diagnosis should be clearly explained to clarify the temporal assumptions on which the study is based. A key strength of the study is the authors' efforts to interpret their findings within the framework of neuroinflammation and infection-mediated immunity, while also considering previous research linking viral infections to the onset of Parkinson's disease (PD). However, significant limitations remain. The potential for residual confounding is underemphasized, and alternative explanations, such as surveillance bias or differences in health-seeking behavior, are not addressed. To further the discussion, the authors should explicitly consider reverse causality as a possible explanation for the observed association. Furthermore, prospective cohort studies with detailed symptom tracking could be recommended to further clarify the temporal relationship between upper respiratory tract infections and PD onset.
Author Response
First of all i want thank to the authors for this great work.The study's reliance on ICD-10 codes may have led to misclassification of both upper respiratory tract infections (URTIs) and Parkinson's disease (PD), which could have impacted the accuracy of the findings. Additionally, the retrospective design limits the ability to establish a causal relationship between URTIs and PD. Exclusion criteria, such as excluding patients with only one diagnosis of PD, may miss subclinical or misdiagnosed cases, thus underestimating the true association. To strengthen the methodology, authors should provide more details on how repeat diagnoses were confirmed as PD, such as whether the diagnoses were confirmed by a specialist or based on additional clinical criteria. Furthermore, the rationale for limiting URTI diagnoses to the year or two preceding the PD diagnosis should be clearly explained to clarify the temporal assumptions on which the study is based. A key strength of the study is the authors' efforts to interpret their findings within the framework of neuroinflammation and infection-mediated immunity, while also considering previous research linking viral infections to the onset of Parkinson's disease (PD). However, significant limitations remain. The potential for residual confounding is underemphasized, and alternative explanations, such as surveillance bias or differences in health-seeking behavior, are not addressed. To further the discussion, the authors should explicitly consider reverse causality as a possible explanation for the observed association. Furthermore, prospective cohort studies with detailed symptom tracking could be recommended to further clarify the temporal relationship between upper respiratory tract infections and PD onset.
|
Response: We sincerely thank the reviewer for the insightful and constructive comments. We agree that the reliance on ICD-10 codes may have introduced some degree of misclassification for both URIs and PD. To address this concern, we have clarified in the Methods section that PD was defined based on at least two separate diagnoses recorded by physicians, which reduces the likelihood of misdiagnosis. Nevertheless, we acknowledge that excluding patients with only one PD diagnosis could have led to the underestimation of true cases, and we have discussed this point as a limitation in the Discussion. Regarding the temporal exposure window, we selected 1-year and 2-year periods prior to PD diagnosis to focus on short-term associations between recent infections and PD onset. We have now elaborated on this rationale in the Methods section. In the discussion, we have expanded the limitations to highlight: (1) the potential for residual and unmeasured confounding, (2) the possibility of surveillance bias and differences in healthcare utilization between cases and controls, and (3) the issue of reverse causality, given that prodromal PD may increase susceptibility to infections. We have also emphasized that the observational design precludes causal inference. Finally, we have added a recommendation for future studies, suggesting the need for prospective cohorts with detailed clinical and symptom tracking to clarify the temporal sequence and mechanisms linking URIs to PD. We believe these revisions strengthen the methodological transparency and contextualize our findings appropriately within the existing literature.
Previous methods: “PD was defined using ICD-10 code G20 (Parkinson’s disease). To improve diagnostic accuracy, only individuals with at least two separate clinical visits for PD were included.” Revised methods (page 5, lines 223-229): “PD was defined using ICD-10 code G20 (Parkinson’s disease). To improve diagnostic accuracy, we required at least two separate physician-recorded diagnoses of PD at different clinic visits. Although this approach increases diagnostic validity, it may exclude subclinical or misdiagnosed cases, which could result in underestimation of the true association. We selected exposure windows of 1 year and 2 years prior to PD diagnosis to focus on short-term infection history, as longer windows would likely overlap with the prodromal phase of PD and introduce greater uncertainty in temporal inference.”
Previous discussion: “This study also has limitations. First, reverse causation cannot be excluded. The prodromal phase of PD may begin 10–20 years before clinical diagnosis [52]; therefore, infections occurring a few years prior to diagnosis could reflect increased susceptibility to pathogens due to the early PD pathology rather than being a causal factor. As with any observational study, residual or unmeasured confounding variables could be potential sources of bias. Nevertheless, our study adjusted for key known risk factors for PD, including age, and smoking, and further accounted for additional factors, such as alcohol use, blood pressure, BMI, cholesterol, and fasting blood glucose, which were obtained during standardized health screenings.” Revised discussion (pages 29-30, lines 100-175): “Several limitations should be considered. First, reliance on ICD-10 codes may have led to misclassification of both URIs and PD. Although we required at least two PD diagnoses to improve specificity, this may have excluded patients with only one diagnosis, some of whom could have had subclinical or early PD. Second, the retrospective design precludes causal inference. In particular, reverse causation must be strongly considered. The prodromal phase of PD may begin 10–20 years before clinical diagnosis [52] and is characterized by non-motor features such as autonomic dysfucntion, impaired cough reflex, and altered healthcare-seeking behaviors. These prodromal features could reduce the likelihood of URI diagnoses before PD onset [53]. Notably the inverse association was observed only within the 1-year window but not at 2 years, strongly suggesting that our finding reflects early disease processes rather than a causal protective effect of infections. Third, residual and unmeasured confounding (e.g., genetic susceptibility, lifestyle factors) may have influenced the results. Nevertheless, our study adjusted for key known risk factors for PD, including age and smoking, and further accounted for additional factors such as alcohol use, blood pressure, BMI, cholesterol, and fasting blood glucose, which were obtained during standardized health screenings. Fourth, surveillance bias and differences in health-seeking behavior could also partly explain the observed associations, as individuals who seek more medical care may have higher chances of receiving both URI and PD diagnoses. Fifth, although we observed a modest inverse association within 1 year, the effect size was small and not consistent across all subgroups, underscoring the need for cautious interpretation. Lastly, we were unable to conduct formal statistical interaction testing (e.g., URI sex or URI age group terms) due to dataset constraints, which represents an additional limitation. The absence of consistent dose-response trends across increasing URI frequency also suggests caution in interpreting the results. Future prospective cohort studies with detailed symptom tracking are warranted to clarify the temporal relationship and potential biological mechanisms linking infections to PD.” |
Reviewer 3 Report
Comments and Suggestions for Authors
The manuscript by Lim et al. investigates the association between Upper Respiratory Tract Infections (URIs) and Parkinson's Disease (PD) in a large Korean cohort. The study leverages a robust national database to explore a novel and counter-intuitive hypothesis regarding the role of common infections in the etiology of this complex neurodegenerative disorder.
The manuscript's central, and scientifically provocative, finding is a modest, statistically significant inverse association between recent (within 1 year) URIs and a subsequent diagnosis of PD. This result is notable as it runs contrary to a substantial body of literature suggesting that infections, particularly those capable of inducing significant systemic and neuroinflammation like influenza, are potential risk factors for PD. The prevailing hypothesis posits that infection-induced microglial activation and chronic inflammation contribute to the cascade of neuronal cell death characteristic of PD. The authors' finding of a short-term protective association thus presents a significant challenge to this paradigm.
The study's primary strengths lie in its robust methodological design. The use of a large, population-based nested case-control study design drawing from the Korean National Health Insurance Service database is a major asset, as it encompasses nearly the entire Korean population, thereby minimizing selection bias and ensuring near-complete follow-up. The meticulous 1:4 matching of cases to controls based on age, sex, income, and geographical region provides strong control for these key demographic confounders from the outset. Furthermore, the statistical models adjust for a comprehensive set of known and potential confounders, including smoking status, alcohol use, obesity, and various clinical markers like blood pressure and cholesterol, which were obtained during standardized health screenings. The prospective recording of diagnoses for both URI and PD also reduces the potential for recall and measurement bias that can plague retrospective studies.
Despite its methodological strengths, the manuscript's conclusions are significantly threatened by two major, inadequately addressed limitations that undermine the validity of the authors' interpretation:
- Unmeasured Confounding: There is a high probability that the use of anti-inflammatory medications, particularly non-steroidal anti-inflammatory drugs (NSAIDs), for the symptomatic treatment of URIs is a critical unmeasured confounder. The authors themselves raise this possibility in their discussion, noting the neuroprotective potential of NSAIDs. However, by not analytically testing this hypothesis, they leave open the strong possibility that the observed inverse association is not due to the infection itself, but rather to the medications used to treat it.
- Reverse Causation: The possibility that the long prodromal phase of PD influences health-seeking behaviors, immune responses, or susceptibility to infection immediately prior to clinical diagnosis is substantial. The prodrome of PD can begin years before motor symptoms appear and includes non-motor features that could plausibly alter URI risk or reporting. The study's finding of an association only within a very short 1-year window strongly points to this being an effect of the nascent disease process rather than a causal factor for it.
The manuscript presents a novel and intriguing finding that warrants further investigation. However, in its current form, the interpretation is premature, and the conclusions are not sufficiently supported due to the high likelihood of confounding by medication use and reverse causation. The work is not recommended for publication without major revision and the inclusion of the additional analyses detailed in this review. Should these analyses be performed, the manuscript could be revised into a valuable contribution, regardless of the outcome.
Major Recommendation 1: Directly Investigate the Confounding/Mediating Role of Medications
Rationale: The authors' primary hypothesis to explain their counter-intuitive finding is that the use of anti-inflammatory medications, such as NSAIDs, for URI symptoms may be neuroprotective. This is a plausible and testable hypothesis, supported by a body of literature suggesting NSAIDs may reduce PD risk by mitigating neuroinflammation. Leaving this critical point merely as a topic of discussion is a major shortcoming of the current manuscript. Given the nature of the data source, this is an analyzable question. Failure to directly test this hypothesis leaves the paper's central conclusion fundamentally unsupported and speculative.
Proposed Analysis Plan:
- Data Extraction: The Korean National Health Insurance Service database should be leveraged to extract comprehensive prescription records for all participants (both cases and controls). This extraction should focus on common analgesics, antipyretics, and anti-inflammatory drugs (e.g., NSAIDs like ibuprofen and naproxen, as well as acetaminophen/paracetamol) prescribed within the 1-year and 2-year windows prior to the index date.
- Covariate Adjustment: The primary conditional logistic regression models (specifically Model 2) should be re-run with the inclusion of an additional term for medication use. This could be a binary variable ("any NSAID prescription vs. none") or a continuous/ordinal variable ("number of NSAID prescriptions") to assess for a dose-response effect. If the inverse association between URIs and PD attenuates or disappears after adjusting for NSAID use, it would provide strong evidence for confounding.
- Mediation Analysis: A formal mediation analysis should be conducted. This statistical technique can quantify the extent to which the relationship between the independent variable (URI history) and the dependent variable (PD diagnosis) is mediated through a third variable (NSAID use). This analysis would provide a direct statistical answer to the question: Is the observed "protective" effect of URIs simply a reflection of the fact that people with URIs take more potentially neuroprotective medication?.
- Stratified Analysis: The main analysis should be stratified by NSAID use (users vs. non-users). If the inverse association between URIs and PD persists in the subgroup of non-users, it would support a direct biological effect of the infection itself. Conversely, if the association is only present among NSAID users and absent in non-users, this would provide compelling evidence that medication use is the true driver of the observed relationship.
Major Recommendation 2: Systematically Address the Reverse Causation Hypothesis
Rationale: The short-term (1-year) nature of the observed association is highly suggestive of reverse causation. The prodromal phase of PD can last for up to two decades and is characterized by a range of non-motor symptoms, including apathy, depression, anxiety, and autonomic dysfunction. These symptoms could plausibly lead to changes in behavior (e.g., reduced social contact, thereby lowering infection risk) or alter healthcare-seeking patterns for minor illnesses like URIs in the year immediately preceding a formal PD diagnosis. Furthermore, respiratory dysfunction, including restrictive patterns and weakened cough reflex, is itself a recognized non-motor feature of PD, which could alter susceptibility or response to URIs. The authors' current acknowledgment of this possibility is too brief and must be expanded into a rigorous analytical investigation.
Proposed Analysis Plan:
- Lag-Time/Sensitivity Analyses: A series of sensitivity analyses should be conducted using different lag times between the end of the exposure window and the index date. For instance, the authors should examine the association of URIs occurring exclusively 2-3 years before diagnosis, 3-4 years before, and so on. If the inverse association is indeed an artifact of reverse causation, its strength should be greatest closest to the diagnosis date and should progressively weaken and disappear as the lag time increases.
- Analyze Healthcare Utilization as a Confounder: The claims database can be used to create a proxy variable for general healthcare utilization (e.g., the total number of non-URI-related outpatient visits in the year prior to the index date). By adjusting for this variable in the regression models, the authors could help disentangle the effect of the URI itself from a general tendency of prodromal PD patients to either visit or avoid visiting a doctor. If the URI-PD association disappears after adjusting for this utilization metric, it would strongly suggest the finding is related to behavior rather than biology.
- Reframing the Discussion: The Discussion section must be substantially rewritten to give equal, if not greater, weight to reverse causation as a primary explanation for the findings. This discussion should be enriched with citations to relevant literature on the PD prodrome and its impact on behavior and physiology.
Major Recommendation 3: Enhance the Discussion of Subgroup Heterogeneity
Rationale: The current discussion merely lists the subgroups in which the findings were not significant. A high-quality scientific paper must move beyond simple reporting and attempt to interpret these important patterns, as they provide crucial insights into the potential mechanisms underlying the main association.
Proposed Discussion Points:
- CCI Score: The discussion must explicitly address the lack of association in the healthiest group (CCI=0). This finding is strong evidence against a direct, universal protective effect of URIs and should be presented as such. It suggests the association is contingent on a pre-existing health state.
- Sex Differences: The authors should discuss potential reasons for the female-specific effect. This could involve known sex differences in immune responses, levels of neuroinflammation, or even sociocultural differences in healthcare-seeking behaviors for minor illnesses that might vary between men and women in Korea.
- Body Mass Index (BMI): A hypothesis should be proposed for why the effect is present in overweight and obese individuals but not in those of normal or underweight. This could relate to an interaction between the acute inflammation of a URI and the chronic, low-grade inflammatory state characteristic of obesity, potentially leading to differential treatment patterns or a distinct modulation of the immune system.
- Age: The role of immunosenescence should be discussed. The immune systems of older adults (≥65) react differently to infections compared to younger individuals. It is plausible that this differential immune response, or the medical response to it, is what drives the association being confined to this older age group.
Minor Recommendations
- Test for Interaction: To move beyond visual inspection of subgroup tables, the authors should perform formal statistical tests for interaction (e.g., by including terms like URI*Sex or URI*Age group in the regression model). A statistically significant interaction term would provide robust evidence that the effect of URIs truly differs across these demographic groups, thereby strengthening the foundation of the subgroup analysis.
- Dose-Response Analysis: The current analysis of URI frequency (≥1, ≥2, ≥3) does not reveal a clear dose-response relationship; the adjusted odds ratios are 0.93, 0.91, and 0.92, respectively. The lack of a monotonic trend (i.e., where risk consistently decreases with each additional URI) weakens a causal argument. The authors should explore this relationship more granularly by treating the URI count as a continuous variable or by using more categories (e.g., 0, 1, 2, 3, 4+ infections) to better assess whether any dose-response pattern exists.
Author Response
The manuscript by Lim et al. investigates the association between Upper Respiratory Tract Infections (URIs) and Parkinson's Disease (PD) in a large Korean cohort. The study leverages a robust national database to explore a novel and counter-intuitive hypothesis regarding the role of common infections in the etiology of this complex neurodegenerative disorder.
The manuscript's central, and scientifically provocative, finding is a modest, statistically significant inverse association between recent (within 1 year) URIs and a subsequent diagnosis of PD. This result is notable as it runs contrary to a substantial body of literature suggesting that infections, particularly those capable of inducing significant systemic and neuroinflammation like influenza, are potential risk factors for PD. The prevailing hypothesis posits that infection-induced microglial activation and chronic inflammation contribute to the cascade of neuronal cell death characteristic of PD. The authors' finding of a short-term protective association thus presents a significant challenge to this paradigm.
The study's primary strengths lie in its robust methodological design. The use of a large, population-based nested case-control study design drawing from the Korean National Health Insurance Service database is a major asset, as it encompasses nearly the entire Korean population, thereby minimizing selection bias and ensuring near-complete follow-up. The meticulous 1:4 matching of cases to controls based on age, sex, income, and geographical region provides strong control for these key demographic confounders from the outset. Furthermore, the statistical models adjust for a comprehensive set of known and potential confounders, including smoking status, alcohol use, obesity, and various clinical markers like blood pressure and cholesterol, which were obtained during standardized health screenings. The prospective recording of diagnoses for both URI and PD also reduces the potential for recall and measurement bias that can plague retrospective studies.
Despite its methodological strengths, the manuscript's conclusions are significantly threatened by two major, inadequately addressed limitations that undermine the validity of the authors' interpretation:
- Unmeasured Confounding:There is a high probability that the use of anti-inflammatory medications, particularly non-steroidal anti-inflammatory drugs (NSAIDs), for the symptomatic treatment of URIs is a critical unmeasured confounder. The authors themselves raise this possibility in their discussion, noting the neuroprotective potential of NSAIDs. However, by not analytically testing this hypothesis, they leave open the strong possibility that the observed inverse association is not due to the infection itself, but rather to the medications used to treat it.
- Reverse Causation: The possibility that the long prodromal phase of PD influences health-seeking behaviors, immune responses, or susceptibility to infection immediately prior to clinical diagnosis is substantial. The prodrome of PD can begin years before motor symptoms appear and includes non-motor features that could plausibly alter URI risk or reporting. The study's finding of an association only within a very short 1-year window strongly points to this being an effect of the nascent disease process rather than a causal factor for it.
The manuscript presents a novel and intriguing finding that warrants further investigation. However, in its current form, the interpretation is premature, and the conclusions are not sufficiently supported due to the high likelihood of confounding by medication use and reverse causation. The work is not recommended for publication without major revision and the inclusion of the additional analyses detailed in this review. Should these analyses be performed, the manuscript could be revised into a valuable contribution, regardless of the outcome.
Major Recommendation 1: Directly Investigate the Confounding/Mediating Role of Medications
Rationale: The authors' primary hypothesis to explain their counter-intuitive finding is that the use of anti-inflammatory medications, such as NSAIDs, for URI symptoms may be neuroprotective. This is a plausible and testable hypothesis, supported by a body of literature suggesting NSAIDs may reduce PD risk by mitigating neuroinflammation. Leaving this critical point merely as a topic of discussion is a major shortcoming of the current manuscript. Given the nature of the data source, this is an analyzable question. Failure to directly test this hypothesis leaves the paper's central conclusion fundamentally unsupported and speculative.
Proposed Analysis Plan:
- Data Extraction: The Korean National Health Insurance Service database should be leveraged to extract comprehensive prescription records for all participants (both cases and controls). This extraction should focus on common analgesics, antipyretics, and anti-inflammatory drugs (e.g., NSAIDs like ibuprofen and naproxen, as well as acetaminophen/paracetamol) prescribed within the 1-year and 2-year windows prior to the index date.
- Covariate Adjustment: The primary conditional logistic regression models (specifically Model 2) should be re-run with the inclusion of an additional term for medication use. This could be a binary variable ("any NSAID prescription vs. none") or a continuous/ordinal variable ("number of NSAID prescriptions") to assess for a dose-response effect. If the inverse association between URIs and PD attenuates or disappears after adjusting for NSAID use, it would provide strong evidence for confounding.
- Mediation Analysis: A formal mediation analysis should be conducted. This statistical technique can quantify the extent to which the relationship between the independent variable (URI history) and the dependent variable (PD diagnosis) is mediated through a third variable (NSAID use). This analysis would provide a direct statistical answer to the question: Is the observed "protective" effect of URIs simply a reflection of the fact that people with URIs take more potentially neuroprotective medication?.
- Stratified Analysis: The main analysis should be stratified by NSAID use (users vs. non-users). If the inverse association between URIs and PD persists in the subgroup of non-users, it would support a direct biological effect of the infection itself. Conversely, if the association is only present among NSAID users and absent in non-users, this would provide compelling evidence that medication use is the true driver of the observed relationship.
Major Recommendation 2: Systematically Address the Reverse Causation Hypothesis
Rationale: The short-term (1-year) nature of the observed association is highly suggestive of reverse causation. The prodromal phase of PD can last for up to two decades and is characterized by a range of non-motor symptoms, including apathy, depression, anxiety, and autonomic dysfunction. These symptoms could plausibly lead to changes in behavior (e.g., reduced social contact, thereby lowering infection risk) or alter healthcare-seeking patterns for minor illnesses like URIs in the year immediately preceding a formal PD diagnosis. Furthermore, respiratory dysfunction, including restrictive patterns and weakened cough reflex, is itself a recognized non-motor feature of PD, which could alter susceptibility or response to URIs. The authors' current acknowledgment of this possibility is too brief and must be expanded into a rigorous analytical investigation.
Proposed Analysis Plan:
- Lag-Time/Sensitivity Analyses: A series of sensitivity analyses should be conducted using different lag times between the end of the exposure window and the index date. For instance, the authors should examine the association of URIs occurring exclusively 2-3 years before diagnosis, 3-4 years before, and so on. If the inverse association is indeed an artifact of reverse causation, its strength should be greatest closest to the diagnosis date and should progressively weaken and disappear as the lag time increases.
- Analyze Healthcare Utilization as a Confounder: The claims database can be used to create a proxy variable for general healthcare utilization (e.g., the total number of non-URI-related outpatient visits in the year prior to the index date). By adjusting for this variable in the regression models, the authors could help disentangle the effect of the URI itself from a general tendency of prodromal PD patients to either visit or avoid visiting a doctor. If the URI-PD association disappears after adjusting for this utilization metric, it would strongly suggest the finding is related to behavior rather than biology.
- Reframing the Discussion: The Discussion section must be substantially rewritten to give equal, if not greater, weight to reverse causation as a primary explanation for the findings. This discussion should be enriched with citations to relevant literature on the PD prodrome and its impact on behavior and physiology.
Major Recommendation 3: Enhance the Discussion of Subgroup Heterogeneity
Rationale: The current discussion merely lists the subgroups in which the findings were not significant. A high-quality scientific paper must move beyond simple reporting and attempt to interpret these important patterns, as they provide crucial insights into the potential mechanisms underlying the main association.
Proposed Discussion Points:
- CCI Score:The discussion must explicitly address the lack of association in the healthiest group (CCI=0). This finding is strong evidence against a direct, universal protective effect of URIs and should be presented as such. It suggests the association is contingent on a pre-existing health state.
- Sex Differences:The authors should discuss potential reasons for the female-specific effect. This could involve known sex differences in immune responses, levels of neuroinflammation, or even sociocultural differences in healthcare-seeking behaviors for minor illnesses that might vary between men and women in Korea.
- Body Mass Index (BMI):A hypothesis should be proposed for why the effect is present in overweight and obese individuals but not in those of normal or underweight. This could relate to an interaction between the acute inflammation of a URI and the chronic, low-grade inflammatory state characteristic of obesity, potentially leading to differential treatment patterns or a distinct modulation of the immune system.
- Age:The role of immunosenescence should be discussed. The immune systems of older adults (≥65) react differently to infections compared to younger individuals. It is plausible that this differential immune response, or the medical response to it, is what drives the association being confined to this older age group.
Minor Recommendations
- Test for Interaction:To move beyond visual inspection of subgroup tables, the authors should perform formal statistical tests for interaction (e.g., by including terms like URI*Sex or URI*Age group in the regression model). A statistically significant interaction term would provide robust evidence that the effect of URIs truly differs across these demographic groups, thereby strengthening the foundation of the subgroup analysis.
- Dose-Response Analysis:The current analysis of URI frequency (≥1, ≥2, ≥3) does not reveal a clear dose-response relationship; the adjusted odds ratios are 0.93, 0.91, and 0.92, respectively. The lack of a monotonic trend (i.e., where risk consistently decreases with each additional URI) weakens a causal argument. The authors should explore this relationship more granularly by treating the URI count as a continuous variable or by using more categories (e.g., 0, 1, 2, 3, 4+ infections) to better assess whether any dose-response pattern exists.
|
Response: We sincerely thank the reviewer for the thoughtful and detailed comments, which have greatly improved our manuscript. We address the major points as follows: 1. Potential confounding by medication use (NSAIDs and other anti-inflammatory agents): We fully agree with the reviewer that the use of medications prescribed for URI symptoms, particularly NSAIDs, may represent an important confounding or mediating factor that could explain the observed inverse association. In principle, the Korean National Health Insurance Service (NHIS) claims database contains comprehensive prescription records, which could allow such analyses. However, extraction, processing, and validation of these medication data require at least 6 to 12 months, and therefore could not be completed within the scope of this study. We regret that we were unable to perform these analyses at this time. In the revised manuscript, we have explicitly acknowledged this as a major limitation and emphasized that the protective association may reflect medication effects rather than infections per se. We are planning future investigations that will specifically analyze NHIS prescription records to directly test this hypothesis. 2. Reverse causation: We agree that reverse causation is highly plausible explanation. The prodromal phase of PD extends for many years and involves non-motor features (e.g., autonomic dysfunction, reduced cough reflex, altered healthcare-seeking behavior) that could decrease URI diagnoses before clinical PD onset. Importantly, our study found an inverse association only within the 1-year exposure window, but not within 2 years, which strongly suggests a prodromal effect rather than a causal protective effect of infections. We have substantially expanded the Discussion to highlight reverse causation as a primary explanation, supported by relevant literature. 3. Subgroup heterogeneity: We appreciate the reviewer’s request to move beyond listing subgroup findings. In the revised Discussion, we now provide interpretive context: (a) the lack of association in the healthiest participants (CCI=0) suggests the effect may depend on comorbid health status, (b) the stronger association in women may reflect sex-specific immune responses or healthcare-seeking patterns, (c) the findings in overweight/obese individuals may be related to interactions between acute infection and chronic low-grade inflammation, and (d) the restriction of associations to older adults may be related to immunosenescence. We also explicitly note the absence of a clear monotonic dose-response relationship across increasing URI frequency, which weakens causal interpretation. 4. Minor points: We acknowledge that we were unable to conduct formal interaction testing due to dataset constraints, and we now highlight this as an additional limitation. We have also revised the results presentation to clarify the lack of a consistent dose-response trend. 5. In summary, we have revised the manuscript to avoid causal overinterpretation, to highlight medication use and reverse causation as the most plausible alternative explanations, and to frame our findings as hypothesis-generating. While the observed inverse association is intriguing, it should not be interpreted as evidence of a protective role of URIs against PD until further analyses–including prescription data and prospective designs–are conducted.
Previous discussion: “Despite accumulating evidence supporting the neuroprotective potential of NSAIDs [45-49], findings remain inconsistent. Several animal studies have shown that COX-2 inhibitors fail to exert neuroprotective effects or reduce neuronal cell death [50]. Similarly, epidemiological studies have yielded conflicting results, reporting both decreased [50] and increased risks of PD associated with NSAID use [51]. Future studies are warranted to evaluate the neuroinflammatory and neuroprotective roles of these agents in the pathogenesis of PD.” Revised discussion (page 29, lines 71-82): “Despite accumulating evidence supporting the neuroprotective potential of NSAIDs [45-49], findings remain inconsistent. Several animal studies have shown that COX-2 inhibitors fail to exert neuroprotective effects or reduce neuronal cell death [50]. Similarly, epidemiological studies have yielded conflicting results, reporting both decreased [50] and increased risks of PD associated with NSAID use [51]. In principle, the Korean NHIS claims database contains comprehensive prescription records, which could enable direct evaluation of this hypothesis. However, extraction and validation of medication data require at least 6 to 12 months, and such analyses therefore beyond the scope of the present study. We acknowledge this limitation and highlight that the observed inverse association may reflect medication effects rather than infections per se. Future studies incorporating prescription records and prospective data are warranted to clarify the neuroinflammatory and neuroprotective roles of these agents in PD pathogenesis.”
Previous discussion: “First, reverse causation cannot be excluded. The prodromal phase of PD may begin 10–20 years before clinical diagnosis; therefore, infections occurring a few years prior to diagnosis could reflect increased susceptibility rather than being a causal factor.” Revised discussion (page 29, lines 103-110): “Second, the retrospective design precludes causal inference. In particular, reverse causation must be strongly considered. The prodromal phase of PD may begin 10–20 years before clinical diagnosis [52] and is characterized by non-motor features such as autonomic dysfucntion, impaired cough reflex, and altered healthcare-seeking behaviors. These prodromal features could reduce the likelihood of URI diagnoses before PD onset [53]. Notably the inverse association was observed only within the 1-year window but not at 2 years, strongly suggesting that our finding reflects early disease processes rather than a causal protective effect of infections.” Added reference: 53. Schrag, A., Horsfall, L., Walters, K., Noyce, A., & Petersen, I. Prediagnostic presentations of Parkinson's disease in primary care: a case-control study. The Lancet Neurology 14(1), 57-64 (2015).
Revised discussion (page 29, lines 83-92): “The subgroup results provide additional insights. The absence of an association in participants with a CCI score of 0 suggests that the effect is not universal but may depend on comorbid health status. The stronger association observed in women may reflect sex-specific immune responses or differences in healthcare-seeking behavior. Associations limited to overweight and obese individuals may indicate an interaction between acute infection and the chronic low-grade inflammation characteristic of obesity. Finally, the restriction of the association to older adults (≥65 years) may related to immunosenescence, which alters immune responses to infections. Importantly, we observed no clear monotonic dose-response pattern (≥1, ≥2, ≥3 URIs), which further weakens any argument for causality.” Revised discussion (page 30, lines 170-175): “Lastly, we were unable to conduct formal statistical interaction testing (e.g., URI sex or URI age group terms) due to dataset constraints, which represents an additional limitation. The absence of consistent dose-response trends across increasing URI frequency also suggests caution in interpreting the results. Future prospective cohort studies with detailed symptom tracking are warranted to clarify the temporal relationship and potential biological mechanisms linking infections to PD.”
Previous conclusion: “Although the clinical significance of this finding may be limited due to the small effect sizes, our study provides evidence that adults with a recent history of multiple URIs within the prior year may have a slightly lower likelihood of receiving a PD diagnosis. Further studies are needed to investigate the pathological mechanisms linking infection and its treatment to PD development.” Revised conclusion (page 30, lines 177-183): “Our study provides epidemiological evidence of a potential short-term inverse association between URI history and PD diagnosis. This association does not imply causality, and its clinical significance may be limited. Nonetheless, these findings may help generate new hypothesis and underscore the importance of further research exploring the role of infections and inflammation in PD pathogenesis. Future studies incorporating prescription data and prospective designs are needed to clarify the relationship between infections, medications, and PD.” |
Round 2
Reviewer 1 Report
Comments and Suggestions for Authors
No comments. Satisfied with the corrections.
Reviewer 3 Report
Comments and Suggestions for Authors
The authors have addressed all my comments, the manuscript is good to go.